# LSRS: Latent Scale Rejection Sampling for Visual Autoregressive Modeling

## Abstract

Visual Autoregressive (VAR) modeling approach for image generation proposes autoregressive processing across hierarchical scales, decoding multiple tokens per scale in parallel. This method achieves high-quality generation while accelerating synthesis. However, parallel token sampling within a scale may lead to structural errors, resulting in suboptimal generated images. To mitigate this, we propose *Latent Scale Rejection Sampling (LSRS)*, a method that progressively refines token maps in the latent scale during inference to enhance VAR models. Our method uses a lightweight scoring model to evaluate multiple candidate token maps sampled at each scale, selecting the high-quality map to guide subsequent scale generation. By prioritizing early scales critical for structural coherence, LSRS effectively mitigates autoregressive error accumulation while maintaining computational efficiency. Experiments demonstrate that LSRS significantly improves VAR's generation quality with minimal additional computational overhead. For the VAR-$d$30 model, LSRS increases the inference time by merely **1%** while reducing its FID score from **1.95** to **1.78**. When the inference time is increased by **15%**, the FID score can be further reduced to **1.66**. LSRS offers an efficient test-time scaling solution for enhancing VAR-based generation. The core code is available at `https://anonymous.4open.science/r/LSRS_anonymous-E2DE`.

## 1 Introduction

Autoregressive generative models predict the next element based on previously generated elements, thereby progressively constructing the entire sequence. In the field of Natural Language Processing, Large Language Models (LLMs) (Brown et al., 2020; OpenAI, 2023; Liu et al., 2024a; Touvron et al., 2023; Bai et al., 2023) have demonstrated the effectiveness of the autoregressive paradigm for text generation. In recent years, numerous studies have begun training large autoregressive models for image generation. These models employ visual tokenizers (Van Den Oord et al., 2017; Razavi et al., 2019; Esser et al., 2021) to discretize images into token sequences, aligning their input format with that of LLMs. Consequently, the training methodologies used for LLMs can be directly applied to autoregressive image models. Visual Autoregressive modeling (VAR) (Tian et al., 2024) proposes a novel paradigm for image generation by replacing the traditional "next-token prediction" with "next-scale prediction". The core idea of VAR is to decompose an image into a sequence of latent scales, where each scale corresponds to a specific resolution and is represented as a token map containing multiple tokens. The scale sequence is arranged in ascending order of resolution. VAR performs autoregression between scales while generating tokens within each scale in parallel. Finally, VAR upsamples and fuses the outputs from all scales, generating the final image through a VQ-VAE (Van Den Oord et al., 2017; Razavi et al., 2019) decoder.

Although VAR (Tian et al., 2024) significantly reduces the number of autoregressive steps and enhances generation speed through parallel in-scale token generation, it exhibits theoretical limitations. The parallel generation essentially performs independent sampling for each token, leading VAR to treat the probability of each token map as the product of all individual token probabilities. This approach is unreasonable and may result in structural errors in the images generated by VAR.

To mitigate the aforementioned issues, we propose *Latent Scale Rejection Sampling (LSRS)*, a lightweight approach that significantly enhances the generation quality of VAR models while maintaining inference efficiency. The core idea of LSRS is to employ rejection sampling at the latent

scale, progressively optimizing the token map at each scale during inference. It utilizes a lightweight scoring network to evaluate and select the high-quality token map for each latent scale. By operating in the latent space and leveraging VAR's inherent property of parallel generation within scales, it introduces minimal additional computational overhead. Experimental results demonstrate that LSRS significantly improves the generation quality of VAR models with minimal overhead. For the VAR-$d$30 model, LSRS increases the inference time by merely **1%** while reducing its FID score from **1.95** to **1.78**. When the inference time is increased by **15%**, the FID score can be further reduced to **1.66**. In summary, our contributions to the community are as follows:

- An analysis of the mechanisms and inherent limitations of VAR. Its independent token sampling within scales may lead to erroneous spatial structures in images.
- We propose *Latent Scale Rejection Sampling (LSRS)*, a novel test-time scaling scheme that optimizes VAR inference in the latent space to mitigate the aforementioned limitations.
- Extensive experiments validate the effectiveness of LSRS on VAR and its variants. LSRS effectively reduces erroneous image structures and enhances the generation quality of VAR, while introducing only minimal additional overhead.

## 2 RELATED WORK

### 2.1 AUTOREGRESSIVE IMAGE GENERATION

These models first represent images as discrete visual tokens, then progressively predict these tokens in a specific order using autoregressive models, with a decoder generating images from the predicted tokens. In early works (Van Den Oord et al., 2016a;b; Chen et al., 2020a; Salimans et al., 2017; Reed et al., 2016; 2017; Chen et al., 2020b), the tokens were simply image pixels, and the generation order followed a raster scan sequence. Regarding token types, VQVAE (Van Den Oord et al., 2017; Razavi et al., 2019) and VQGAN (Esser et al., 2021) improved upon this by using feature vectors from the encoder as tokens. In terms of model architecture, LlamaGEN (Sun et al., 2024) and Lumina-mGPT (Liu et al., 2024b) employ GPT-style models (Brown et al., 2020; OpenAI, 2023; Vaswani et al., 2017) for autoregressive modeling. AiM (Li et al., 2024a) and MARS (He et al., 2025a) introduce mixture-of-experts (Jacobs et al., 1991) systems combined with linear attention mechanisms (Gu & Dao, 2023). MaskGIT (Chang et al., 2022) employs bidirectional attention for generation. Methods like SHOW-O (Xie et al., 2024), Transfusion (Zhou et al., 2024), HART (Tang et al., 2024), ResGen (Kim et al., 2024) and DART (Gu et al., 2024) further integrate diffusion models (Sohl-Dickstein et al., 2015; Ho et al., 2020) into autoregressive frameworks. Numerous works have also explored different autoregressive ordering strategies: VAR (Tian et al., 2024), Infinity (Han et al., 2024), and FlexVAR (Jiao et al., 2025) use a latent scale progression from small to large, while RandAR (Pang et al., 2024), RAR (Yu et al., 2024a), SAR (Liu et al., 2024c), MAR (Li et al., 2024b), and ARPG (Li et al., 2025) employ random ordering over token sets. CTF (Guo et al., 2025) proposes a coarse-to-fine approach for token prediction. FAR (Yu et al., 2025) and NFIG (Huang et al., 2025) perform autoregressive modeling in the frequency domain.

### 2.2 REJECTION SAMPLING IN IMAGE GENERATIVE MODELS

Early works (Azadi et al., 2018; Bauer & Mnih, 2019) combine learned rejection sampling schemes with prior distributions to narrow the gap between aggregated posterior distributions. Methods such as VQ-VAE-2 (Van Den Oord et al., 2017), VQGAN (Esser et al., 2021), RQ-VAE (Lee et al., 2022), ViT-VQGAN (Yu et al., 2021), CART (Roheda, 2024) and VAR (Tian et al., 2024) employ traditional rejection sampling approaches. They require the model to generate a complete image first, and then use image classification models like ResNet-101 (He et al., 2016) for screening, which results in extremely low efficiency. DDO (Zheng et al., 2025) fine-tunes the generative model itself to increase the probability of generating high-quality images. Several works use rejection sampling in latent space: Issenhuth et al. (2022) uses a network to perform iterative rejection sampling on the prior distribution until a sample is accepted, which is then passed to a GAN for generation. Che et al. (2020) defines an energy model for latent space sampling. Variational rejection sampling (Grover et al., 2018) integrates rejection sampling into the variational inference of latent variable models to improve accuracy. Dual rejection sampling (Hou et al., 2020) employs a discriminator-based scheme to correct the generative prior in latent space. RS-IMLE (Vashist et al., 2024) modifies the training

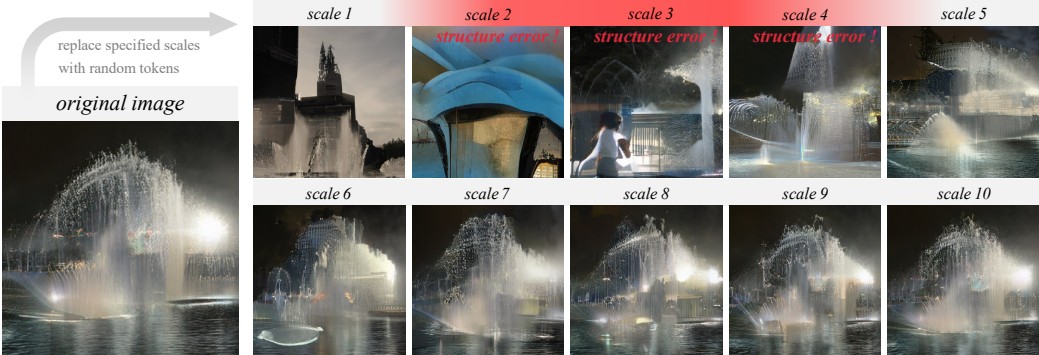

Figure 1: The leftmost image is generated using VAR-$d$30 with the class label "fountain". The images labeled from scale 1 to 10 are obtained by replacing the token maps of VAR at each individual scale with random token maps and then decoding the final images.

prior of IMLE (Li & Malik, 2018) via rejection sampling. Diffusion rejection sampling (Na et al., 2024) determines whether to accept a sampling result based on the ratio of true to model transition kernels. If rejected, it reverts the noise. This leads to substantial additional computation. Wang et al. (2024) proposes continuous speculative decoding for autoregressive image generation models.

## 3 METHOD

### 3.1 PRELIMINARY

Visual Autoregressive (VAR) modeling (Tian et al., 2024) proposes a "next-scale prediction" strategy, where the smallest unit of autoregression is a token map composed of multiple discrete tokens. VAR quantizes the feature map $f \in \mathbb{R}^{H \times W \times C}$ into $K$ multi-scale discrete token maps $(r_1, r_2, \ldots, r_K)$, where $r_k \in \mathbb{Z}^{h_k \times w_k}$ contains $h_k \times w_k$ tokens. The autoregressive likelihood is formulated as:

$$p(r_1, r_2, \ldots, r_K) = \prod_{k=1}^{K} p(r_k \mid r_1, r_2, \ldots, r_{k-1}). \tag{1}$$

During inference, at step $k$, VAR uses $\{r_{<k}\}$ to predict the probability distribution of all tokens in $r_k$. These tokens are then sampled independently. The token maps from all scales are then upsampled to the feature map resolution and fused, before being fed into the decoder to generate the final image. In summary, the VAR model performs autoregressive prediction across scales for token maps, while all tokens within each scale's token map are generated in parallel.

### 3.2 OBSERVATION

**Imperfect parallel sampling mechanism.** Although the VAR model significantly reduces the number of autoregressive steps and enhances generation speed through parallel in-scale token generation, it exhibits theoretical limitations. Specifically, for scales $k$ where the token map $r_k$ contains more than one token (i.e., $r_k$ with $h_k \times w_k > 1$), the VAR forward pass computes the probability distribution over the vocabulary $V$ for each token in $r_k$. Formally, for token map $r_k$, the probability distribution for each token $r_k(i,j)$ at position $(i,j)$ is computed as $p(r_k(i,j) \mid r_{<k})$. Each token $r_k(i,j)$ is then sampled independently from its respective distribution to obtain an index, and these indices are combined to form the sampled token map $r_k$. This means the joint probability of the token map $r_k$ is modeled as the product of the individual token probabilities:

$$p(r_k \mid r_1, r_2, \ldots, r_{k-1}) = \prod_{i=1}^{h_k} \prod_{j=1}^{w_k} p(r_k(i,j) \mid r_1, r_2, \ldots, r_{k-1}). \tag{2}$$

This is incorrect because tokens within the same scale are not mutually independent. In reality, the sampled tokens should influence the distribution of the unsampled tokens, especially for neighboring tokens. However, for VAR, the distributions of all tokens $p(r_k(i,j) \mid r_{<k})$ in $r_k$ are generated in

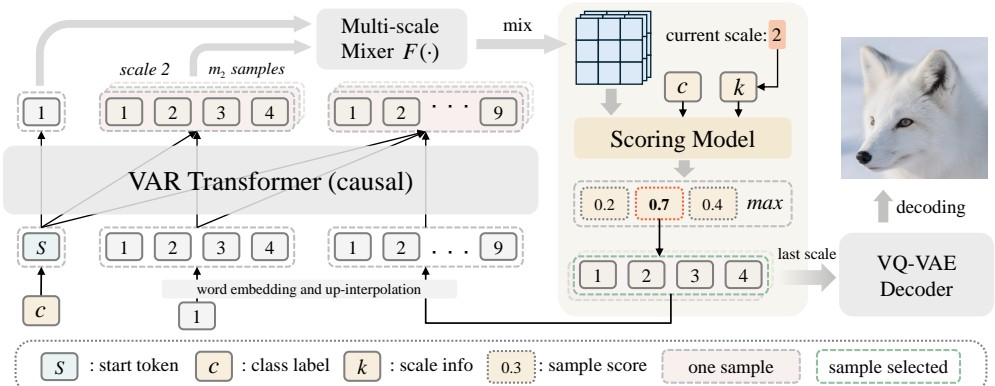

Figure 2: **An illustration of LSRS applied during VAR inference.** At each scale, multiple candidate token maps are sampled from VAR's output distribution. The LSRS scoring model then evaluates each token map, and the one with the highest score is selected as the final output for that scale.

parallel by a Transformer (Vaswani et al., 2017). This means VAR cannot achieve this effect and may introduce errors in cases where multiple tokens exhibit dependencies.

**Earlier scales influence more.** CoDe (Chen et al., 2024) reveals that early scales contain low-frequency information, which is more critical for the generation quality of VAR. To investigate the impact of each scale to the overall image, we replace the predicted token maps with random ones at each scale individually. The decoded images are shown in Figure 1. The image category is "fountain", and it can be observed that earlier scales have a greater impact on the image structure. Random replacement at scales 2, 3 and 4 introduces severe structural errors. This proves that the early stage scale determines the spatial structure of the image. Additionally, due to the scale-wise autoregressive property of VAR, errors in the early stage scale will lead to the accumulation of errors in subsequent scales, ultimately resulting in a distorted image.

Combining the two observations above, if VAR's parallel sampling leads to poor quality at earlier scales, it can easily result in the generation of low-quality or even incorrect images. Therefore, if we can optimize the early scale, we can then efficiently improve the quality of the final images.

### 3.3 LSRS: REJECTION SAMPLING IN LATENT SCALE

Now we need a method superior to Equation 2 for evaluating each $r_k$. Inspired by discriminators in GANs (Goodfellow et al., 2020; Brock et al., 2018; Sauer et al., 2022) and reward models in RLHF (Schulman et al., 2017; Rafailov et al., 2023; Shao et al., 2024; Meng et al., 2024), we employ a scoring model to implicitly capture the dependencies among all tokens within a sample of $r_k$. The model then outputs a scalar score representing the overall quality of the sample. Subsequently, we perform rejection sampling over multiple samples of $r_k$ based on these scores. This constitutes the core idea of *LSRS (Latent Scale Rejection Sampling)*.

Training of the scoring model is grounded in one empirical observation and one assumption: (1) Real images always exhibit correct spatial structure, whereas generated images may contain distortions; (2) The quality of generated images can hardly surpass that of real images. Consequently, the training objective for the scoring model is to assign low scores to generated data and high scores to real data.

**Dataset construction.** To improve efficiency, we construct a static dataset. For each class in the ImageNet-1k dataset, we employ a pre-trained VAR model to generate a large set of images and extract their multi-scale token maps $(r_1^{gen}, r_2^{gen}, \ldots, r_K^{gen})$, where $r_k^{gen} \in \mathbb{Z}^{h_k \times w_k}$ represents the token map at scale $k$ with $h_k \times w_k$ discrete tokens. We construct a data point as $(c, r_k^{gen}, k; 0)$, where $c$ denotes the class label, $k$ represents the scale number, and $0$ indicates that this data point is generated. Similarly, for real images from ImageNet-1k, we apply the Multi-scale VQVAE (Tian et al., 2024) to quantize the feature maps $f \in \mathbb{R}^{H \times W \times C}$ into corresponding multi-scale token maps $(r_1^{real}, r_2^{real}, \ldots, r_K^{real})$, then construct a data point $(c, r_k^{real}, k; 1)$, where $1$ indicates that this data point is from a real image. We save all the data points of all scales for scoring model training.

**Scoring model training.** We use a lightweight neural network $S$ as the scoring model. For $r_k$ from the same image, we define $e_k = F(\{r_{\leq k}\}) \in \mathbb{R}^{H \times W \times C}$, where $F(\cdot)$ denotes the operator in VAR

that fuses token maps from multiple scales into a single feature map at original resolution. The specific details of $F(\cdot)$ are provided in Appendix A. In short, $e_k$ is the feature map obtained by fusing all $r_{\leq k}$. Scoring model takes a triplet $(c, e_k, k)$ as input. The model outputs a scalar score $S(c, e_k, k) \in \mathbb{R}$, reflecting the quality of a token map. The training objective is to assign higher scores to real token maps compared to generated ones, formalized as:

$$S(c, e_k^{\text{real}}, k) > S(c, e_k^{\text{gen}}, k). \tag{3}$$

To enable the model to learn to assign higher scores to real data, the model can be trained using a pairwise log-sigmoid rank loss (Burges et al., 2005), which optimizes the relative ranking of real versus generated token maps:

$$\mathcal{L}_{\text{PairWise}} = - \sum_{(c, e_k^{\text{real}}, e_k^{\text{gen}}, k)} \log \sigma(S(c, e_k^{\text{real}}, k) - S(c, e_k^{\text{gen}}, k)), \tag{4}$$

where $\sigma$ is the sigmoid function. Alternatively, a pointwise binary classification loss can be used, where real token maps are labeled as positive ($y = 1$) and generated token maps as negative ($y = 0$):

$$\mathcal{L}_{\text{PointWise}} = - \sum_{(c, e_k, k; y)} \left[ y \log \sigma(S(c, e_k, k)) + (1 - y) \log(1 - \sigma(S(c, e_k, k))) \right]. \tag{5}$$

**LSRS inference.** During inference, LSRS employs a test-time scaling strategy like Best-of-N parameterized by $\{m_1, m_2, \ldots, m_K\}$, where $m_k$ denotes the number of token maps to sample at scale $k$. The distribution of each scale token map is computed in the same way as in the original VAR, i.e., $p(r_k \mid r_{<k})$. We sample $m_k$ token maps $r_k^{(i)} \sim p(r_k \mid r_{<k})$. Each $e_k^{(i)} = F(r_{<k}, r_k^{(i)})$ is then evaluated by the scoring model. The token map with the highest score is selected as the final token map for scale $k$:

$$r_k = \arg\max_{r_k^{(i)}} S(c, e_k^{(i)}, k). \tag{6}$$

This process is repeated across all scales $k = 1, 2, \ldots, K$ as described in Algorithm 1 and Figure 2. The rationale for using greedy selection is provided in Appendix G. Finally, All the selected token maps are then upsampled to $H \times W$, fused, and passed to the decoder to generate the image.

---

**Algorithm 1** VAR Inference with Latent Scale Rejection Sampling

---

**Require:** Class label $c$, the pre-trained VAR model, scoring model $S$, number of samples per scale $\{m_1, m_2, \ldots, m_K\}$, number of scales $K$
1: Initialize empty set of selected token maps $\{r_{<1}\} = \emptyset$
2: **for** $k = 1$ to $K$ **do** ▷ Iterate over each scale
3:      Compute distribution $p(r_k \mid r_{<k})$ using pre-trained VAR model
4:      Sample $m_k$ token maps $\{r_k^{(i)}\}_{i=1}^{m_k}$ from $p(r_k \mid r_{<k})$, and compute $\{e_k^{(i)} = F(r_{<k}, r_k^{(i)})\}_{i=1}^{m_k}$
5:      Select $r_k = \arg\max_{r_k^{(i)}} S(c, e_k^{(i)}, k)$ ▷ Choose token map with highest score
6:      Update $\{r_{<k+1}\} = \{r_{<k}\} \cup \{r_k\}$
7: **end for**
8: **return** Final token map set $\{r_1, r_2, \ldots, r_K\}$

---

## 4 EXPERIMENTS

**Setup.** All parameters during model sampling remain consistent with the VAR (Tian et al., 2024) setup, specifically with cfg = 1.5, top_p = 0.96, top_k = 900, without using more-smooth, and generating 50 images per class in ImageNet-1k (Deng et al., 2009) for evaluation. We reevaluate the VAR generation metrics for fair comparison, which shows slight differences from the results reported in their original paper. Unless otherwise specified, we employ the pairwise log-sigmoid rank loss for training the scoring model. We construct sampling datasets for each depth of the VAR model separately and pair them with real data. To prevent data leakage, the random seeds used for constructing the VAR sampling dataset are different from those employed during evaluation. For the number of samples at each scale $\{m_1, m_2, \ldots, m_K\}$, we simplify them as $ST$ and $M$ for experimental convenience. $ST$ denotes that scales $ST \sim K$ utilize LSRS while scales $1 \sim ST - 1$ do not. $M$ represents the number of token maps sampled at each scale where LSRS is applied. The implementation details regarding the dataset and scoring model can be found in Appendix A.

Table 1: **Generative model comparison on class-conditional ImageNet (Deng et al., 2009) 256×256**. Metrics include Fréchet inception distance (FID), inception score (IS), precision (Pre) and recall (rec). Step: the number of model runs needed to generate an image. Time: the relative inference time of VAR-$d$30. LSRS is applied to VAR models at various depths, achieving improvements across all cases. Due to space constraints, we only list the methods with FID < 3.

| Model | FID↓ | IS↑ | Pre↑ | Rec↑ | Param | Step | Time |
|---|---|---|---|---|---|---|---|
| StyleGan-XL (Sauer et al., 2022) | 2.30 | 265.1 | 0.78 | 0.53 | 166M | 1 | 0.2 |
| DiT-XL/2 (Peebles & Xie, 2023) | 2.27 | 278.2 | 0.83 | 0.57 | 675M | 250 | 2 |
| MAGVIT-v2 (Yu et al., 2023) | 1.78 | 319.4 | – | – | 307M | 64 | – |
| TiTok-S-128 (Yu et al., 2024b) | 1.97 | 281.8 | – | – | 287M | 256 | 2.21 |
| LlamaGen-XL (Sun et al., 2024) | 2.62 | 244.1 | 0.80 | 0.57 | 775M | 256 | 27 |
| AiM (Li et al., 2024a) | 2.56 | 257.2 | 0.81 | 0.57 | 763M | 256 | 12 |
| SAR-XL (Liu et al., 2024c) | 2.76 | 273.8 | 0.84 | 0.55 | 893M | 256 | – |
| Open-MAGVIT2-XL (Luo et al., 2024) | 2.33 | 271.8 | 0.84 | 0.54 | 1.5B | 256 | – |
| MaskBit (Weber et al., 2024) | 1.52 | 328.6 | – | – | 305M | 256 | 24.3 |
| RAR-XL (Yu et al., 2024a) | 1.50 | 306.9 | 0.80 | 0.62 | 955M | 256 | 2.08 |
| ARPG-XXL (Li et al., 2025) | 1.94 | 339.7 | 0.81 | 0.59 | 1.3B | 64 | 0.97 |
| NAR-XXL (He et al., 2025b) | 2.58 | 293.5 | 0.82 | 0.57 | 1.46B | 31 | 0.53 |
| xAR-H (Ren et al., 2025) | 1.24 | 301.6 | 0.83 | 0.64 | 1.1B | – | 13.0 |
| TokenBridge-H (Wang et al., 2025) | 1.55 | 313.3 | 0.80 | 0.65 | 910M | 256 | 9.55 |
| NFIG (Huang et al., 2025) | 2.81 | 332.42 | 0.77 | 0.59 | 310M | 10 | 0.20 |
| M-VAR-$d$32 (Ren et al., 2024) | 1.78 | 331.2 | 0.83 | 0.61 | 3.0B | 10 | 1.43 |
| MAR-L (Li et al., 2024b) | 1.78 | 296.0 | 0.81 | 0.60 | 479M | 256 | 34.6 |
| MAR-H (Li et al., 2024b) | 1.55 | 303.7 | 0.81 | 0.62 | 943M | 256 | 56.7 |
| RandAR-XL (Pang et al., 2024) | 2.25 | 317.77 | 0.80 | 0.60 | 775M | 88 | 1.47 |
| RandAR-XL (Pang et al., 2024) | 2.22 | 314.21 | 0.80 | 0.60 | 775M | 256 | 4.27 |
| RandAR-XXL (Pang et al., 2024) | 2.15 | 321.9 | 0.79 | 0.62 | 1.4B | 88 | 2.35 |
| FlexVAR-$d$24 (Jiao et al., 2025) | 2.23 | 283.9 | 0.83 | 0.59 | 1.0B | 10 | 0.50 |
| **+ LSRS** $M = 2$ | 2.13 | 284.3 | 0.82 | 0.60 | 1.0B+4M | 10 | 0.51 |
| **+ LSRS** $M = 4$ | 2.09 | 283.4 | 0.82 | 0.60 | 1.0B+4M | 10 | 0.51 |
| VAR-$d$30 (Tian et al., 2024) | 1.95 | 303.1 | 0.82 | 0.59 | 2.0B | 10 | 1.00 |
| **+ LSRS** $M = 4$ | 1.78 | 305.9 | 0.81 | 0.61 | 2.0B+4M | 10 | 1.01 |
| **+ LSRS** $M = 128$ | **1.66** | 298.9 | 0.80 | 0.63 | 2.0B+4M | 10 | 1.15 |

## 4.1 IMAGE GENERATION

**Quality improvement.** The main results are presented in Table 1, where all LSRS configurations employ $ST = 2$. The rationale for this choice will be elaborated in the subsequent ablation studies. As shown in the table, LSRS consistently improves both Fréchet Inception Distance (FID) (Heusel et al., 2017) and Inception Score (IS) (Salimans et al., 2016) across VAR model and its variant FlexVAR. More experiments on VAR can be found in Appendix B. In the largest VAR model VAR-$d$30, LSRS achieves a significant improvement from 1.95 to 1.66 when $M = 128$. These results demonstrate that our proposed LSRS effectively enhances the image generation quality of VAR models.

**Efficiency of LSRS.** Among methods with inference time close to 1, VAR + LSRS achieves the best performance. Notably, LSRS adds only 4M parameters and incurs minimal additional inference time. The performance improvement brought by LSRS is comparable to that of models whose parameter or steps are doubled, such as MAR-L (Li et al., 2024b) and RandAR-XL (Pang et al., 2024). This demonstrates that, compared to enhancing performance by increasing model parameters or sampling steps, LSRS is significantly more efficient.

**Computational cost analysis.** LSRS operates in the latent space and leverages the property of parallel sampling within VAR scales, making its computational cost significantly lower compared to traditional rejection sampling (VAR-$d$30-re). As shown in Table 1, when $M$ is set to a small value ($M = 4$), LSRS introduces almost no additional inference time while still achieving notable FID improvement. More data are presented in Appendix C, demonstrating that LSRS incurs minimal GPU memory overhead. The additional computational time of LSRS increases linearly with $M$, but the absolute increase remains relatively small.

Table 2: Comparison of FID, IS, and sFID metrics for different values of $M$ under pointwise binary classification loss and pairwise log-sigmoid rank loss.

| Inference Model | PointWise loss | | | PairWise loss | | |
|---|---|---|---|---|---|---|
| | FID↓ | IS↑ | sFID↓ | FID↓ | IS↑ | sFID↓ |
| VAR-$d30$ | 1.95 | 303.1 | 8.50 | 1.95 | 303.1 | 8.50 |
| **+ LSRS** $M = 4$ | 1.79 | 305.1 | 7.13 | 1.78 | 305.9 | 7.11 |
| **+ LSRS** $M = 8$ | 1.76 | 303.2 | 6.64 | 1.73 | 303.4 | 6.73 |
| **+ LSRS** $M = 16$ | 1.73 | 302.6 | 6.42 | 1.71 | 302.2 | 6.60 |
| **+ LSRS** $M = 32$ | 1.73 | 303.2 | 6.25 | 1.68 | 300.8 | 6.38 |

Table 3: FID across $M$ Values under $ST = 1$, the first scale with only one token.

| $M$ | $-$ | 2 | 4 | 8 | 16 | 32 | 64 |
|---|---|---|---|---|---|---|---|
| FID | 1.95 | 1.83 ($-0.12$) | 1.78 ($-0.17$) | 1.89 ($-0.06$) | 2.05 ($+0.10$) | 2.30 ($+0.35$) | 2.63 ($+0.68$) |

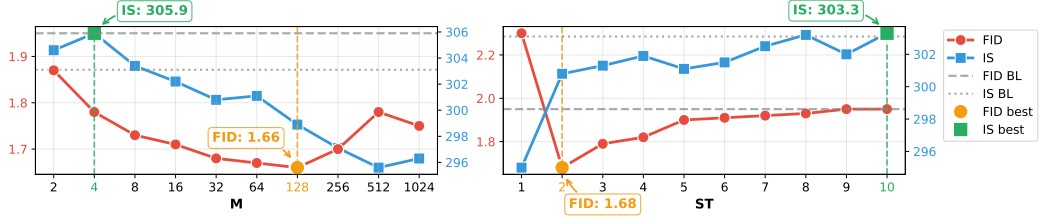

Figure 3: **Ablation experiment on hyperparameter $M$ and $ST$.** Left: Metrics across $M$ values with $ST = 2$. Right: Metrics across $ST$ values with $M = 32$. FID BL and IS BL denote the baseline metrics, i.e., those of the original VAR model. Detailed data can be found in Appendix E.

## 4.2 ABLATION STUDY

**Loss function.** As previously mentioned, the LSRS can utilize both pointwise binary classification loss and pairwise log-sigmoid rank loss. In Table 2, we present the performance of scoring models trained with these two loss functions respectively. Both models are evaluated on VAR-$d30$ with $ST = 2$. The results demonstrate comparable performance between the two approaches, with the pairwise loss showing slightly better FID scores. Consequently, our main experiments are conducted using the scoring model trained with the pairwise loss.

**Which scale to start?** The VAR model employs autoregression across 10 different scales, so theoretically LSRS should perform better when applied to earlier scales. We fixed $M = 32$ and only varied $ST$ on VAR-$d30$ to validate this hypothesis. As shown in the right side of Figure 3, LSRS achieves the optimal FID when $ST = 2$. The FID gradually increases when $ST > 2$ as the control capability of LSRS progressively diminishes.

Notably, if LSRS is applied starting from the first scale (i.e., $ST = 1$), the performance significantly deteriorates compared to the original VAR. Table 3 also demonstrates that when $ST = 1$, the image generation quality tends to deteriorate more easily as $M$ increases. This is expected because the first scale contains only one token and lacks spatial structural information. Moreover, in VAR's multi-scale VQ-VAE, this token is upsampled to the full feature map resolution, effectively acting as a bias term for the entire feature map. We hypothesize that the first scale primarily guides the diversity of generated images. Applying LSRS from the first scale causes many samples to converge to similar values at this scale, thereby reducing generation diversity and degrading FID.

**How many token maps to sample?** As discussed earlier, it is suboptimal to employ LSRS starting from the first scale. Therefore, in this section, we fix $ST = 2$ and only vary $M$, which represents the number of token maps sampled at each scale using LSRS. Theoretically, sampling more token maps increases the likelihood of the model discovering better token maps. In the left side of Figure 3, we present image generation metrics on VAR-$d30$ with $M$ ranging from 2 to 1024. When $M$ takes a relatively small value, it can already yield a considerable improvement in FID. As $M$ increases from 2 to 128, the FID gradually improves. However, further increasing $M$ beyond 128 leads to worse

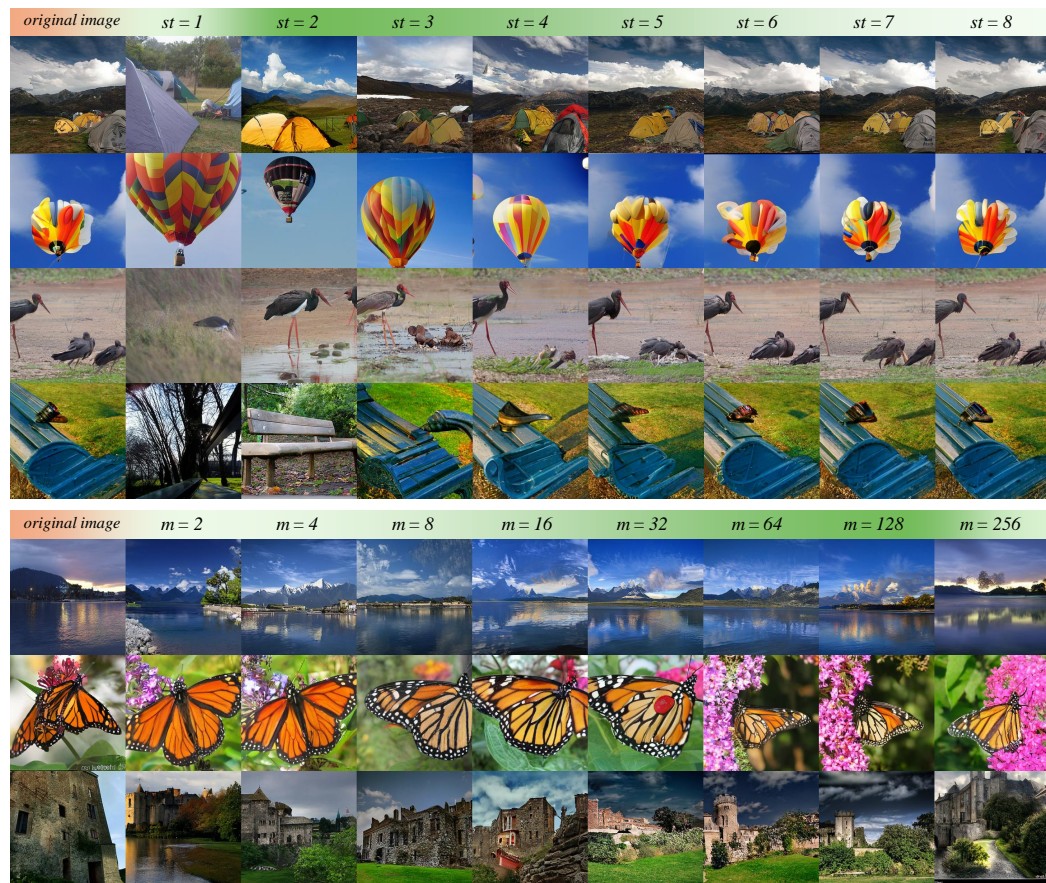

Figure 4: **LSRS Generation Results Demonstration.** The leftmost image is the original VAR-$d30$ generation, while the others show results after LSRS intervention. From top to bottom, they are: mountain tent, balloon, black stork, park bench, lakeside, monarch butterfly and castle.

FID compared to $M = 128$. This may again be attributed to the fact that excessively large $M$ could reduce the diversity of generated images. When $M$ is too large, most of the generated images tend to converge toward a subset defined by the scoring model as the most "safe" but least imaginative. Additionally, the scoring model of LSRS is not perfectly accurate, and larger $M$ may introduce more erroneous scoring of token maps. Additional analysis of the performance decline under large values of $M$ can be found in Appendix F.

**Images generated with LSRS.** In Figure 4, we present the sampled images from the original VAR-$d30$ (leftmost column is the original image generated by VAR) and their counterparts after applying LSRS. Each row corresponds to a specific object category with fixed randomness. The top four rows demonstrate results with fixed LSRS parameter $M = 64$ while varying $ST$, whereas the bottom three rows show cases with fixed $ST = 2$ while varying $M$. It is evident from the first four rows that applying LSRS earlier leads to more pronounced structural corrections in the generated images. Taking the "balloon" image in the second row as an example, the balloon generated by the original VAR exhibits unusual structural errors. The earlier LSRS is applied to the VAR, the greater the correction to the balloon's structure. When $ST \leq 4$, the generated balloon images show no noticeable structural errors. We present additional sampling results in Appendix I, where LSRS effectively improves the suboptimal composition of some images.

It should be particularly noted that the images generated by the VAR model do not always contain structural errors. For example, in the last three rows of Figure 4, the images generated by the original VAR contain no structural errors. However, as $M$ in LSRS increases, the quality of the generated images continues to improve and gradually approaches saturation. In short, the role of LSRS is to correct structural errors as early as possible when VAR samples an incorrect structure, and to further enhance image quality when VAR has already sampled a reasonable structure.

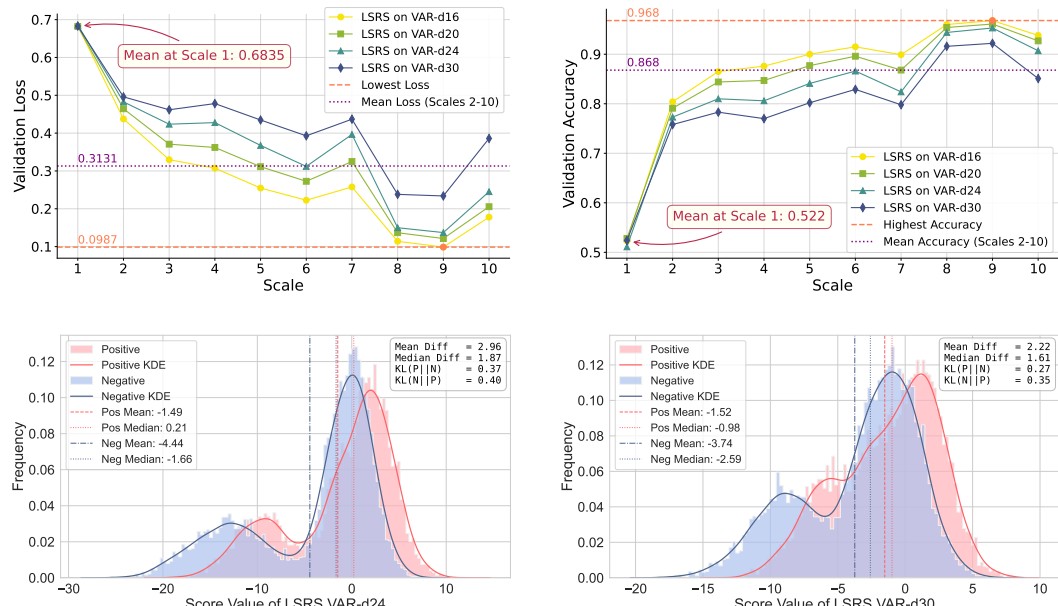

Figure 5: **LSRS scoring model analysis.** Top-left: validation loss. Top-right: validation accuracy. Bottom-left: VAR-$d24$ score distribution. Bottom-right: VAR-$d30$ score distribution. Overall, the accuracy of the scoring model tends to be higher at larger scales and with smaller models.

## 4.3 SCORING MODEL

**Performance at each scale.** As illustrated in Figure 5, the top-left panel displays the loss at each scale on the validation set for scoring models trained on VAR models of varying depths, while the top-right panel presents the corresponding accuracy rates. Since we employ the log-sigmoid rank loss, when the generated data and real data share identical scores, the loss approaches $-\log \text{sigmoid}(0) \approx 0.6931$, rendering the model incapable of distinguishing between them. We observe that the loss for the first scale shows negligible reduction, with the accuracy rate barely exceeding 0.5, indicating the scoring model's inability to assess the authenticity of the first scale. Even so, our supplementary experiments in Appendix H indicate that excluding the first scale during training has no impact on the results. As the scale progresses, the model's accuracy generally exhibits an upward trend. However, with increasing model size, the scoring model's accuracy declines, reflecting a diminishing gap between generated and real data.

**Score distribution.** The left panel in the second row depicts the score distribution output by LSRS trained on VAR-$d24$ on the validation set, while the right panel shows that of VAR-$d30$. These scores exclude the first scale due to the model's aforementioned inaccuracy in judging it. Blue corresponds to generated data (negative samples), and red represents real data (positive samples). The scores of generated data are generally lower than those of real data. The mean score difference between real and generated data for VAR-$d30$ is 2.22, smaller than VAR-$d24$'s 2.96. The KL divergence between the distributions of generated data and real data scores is also smaller for VAR-$d30$. These statistical results are consistent with the observed trend that the accuracy of the scoring model decreases as model size increases. The score distributions of LSRS trained on VAR-$d16$ and VAR-$d20$ are provided in Appendix D. Their score difference between real and generated data is relatively larger.

## 5 CONCLUSION

In this paper, we first reveal the inherent limitations of Visual Autoregressive Modeling (VAR), which could lead to structural errors in generated images. To address this issue, we propose *Latent Scale Rejection Sampling (LSRS)*, an effective and efficient method for improving the generation quality of VAR models. Experimental results demonstrate that LSRS can significantly enhance VAR model performance with minimal additional parameters and overhead. As a novel test-time scaling strategy specifically designed for VAR models, LSRS achieves an exceptional trade-off between efficiency and quality, setting a new benchmark for efficient high-quality image generation.

ETHICS STATEMENT

We adhere to the ICLR Code of Ethics. Our work introduces a foundational technical improvement for Visual Autoregressive (VAR) models, and we have considered the broader ethical implications of our research. Our model is trained on the ImageNet-1k dataset and may inherit its known societal biases. This limitation should be considered in any potential application. Furthermore, like all high-quality image generation technologies, our method carries a risk of misuse for creating deceptive synthetic media. We support community efforts to develop robust detection methods and responsible deployment guidelines. To ensure research integrity and reproducibility, we have made our source code publicly available. This work does not involve human subjects or personally identifiable data.

REPRODUCIBILITY STATEMENT

To support the reproducibility of our research, we have provided an anonymous link to the core source code in the abstract: `https://anonymous.4open.science/r/LSRS_ anonymous-E2DE`. This code repository contains a complete implementation of our proposed method, including training and inference scripts, as well as configuration files. All preprocessing steps and hyperparameter settings are documented within the repository files and in Appendix A. We encourage readers to consult this code repository to access the full technical details necessary for reproducing our results.

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

# A   IMPLEMENTATION DETAILS

**Dataset Construction.** We use the official VAR model weights, code, and sample settings (cfg = 1.5, top_p = 0.96, top_k = 900, without using more-smooth) to sample each of the 1,000 classes in the ImageNet-1k (Deng et al., 2009) dataset 1,000 times, saving the index IDs of the token maps. Sampling 1,000 times roughly aligns with the order of magnitude of images per class in ImageNet-1k, which facilitates training the subsequent scoring model.

**Scoring Model.** To minimize the additional computational overhead during LSRS inference, we design a lightweight convolutional neural network (LeCun et al., 1998; 1989) as the scoring model. Its core consists of multiple residual convolutional blocks where each residual block comprises three convolutional layers (3x3, 3x3, 1x1), LeakyReLU (Maas et al., 2013; Nair & Hinton, 2010), LayerNorm-2d (Ba et al., 2016) and residual connections (He et al., 2016; Srivastava et al., 2015).

The operator $F(\cdot)$ originates from the multi-scale VQVAE of VAR (Tian et al., 2024). It maps each input token map through the codebook's embedding layer (Bengio et al., 2003) to obtain feature maps at their respective scales. These feature maps are then upsampled to the original latent space size $H \times W$ and summed together to produce the final feature map. The final feature map serves as input to the scoring model network. After passing through several residual convolutions, they are transformed into $256 \times 2 \times 2$ visual features, which are then flattened into a 1024-dimensional vector. Subsequently, the class labels (1000 categories) and scale information (10 categories) are mapped to 128-dimensional vectors via their respective embedding layers (Bengio et al., 2003). The visual features are concatenated with the class and scale embeddings, and the combined representation is fed into a multi-layer MLP for fusion, ultimately producing a scalar score.

The training is conducted for 4 epochs with a default batch size of 128 and a learning rate of $3 \times 10^{-4}$. The Adam optimizer (Kingma, 2014) is employed, along with a learning rate scheduling strategy consisting of linear warmup (Goyal et al., 2017) (1 epoch) followed by cosine decay (Loshchilov & Hutter, 2016) (remaining epochs).

**LSRS for FlexVAR.** For the FlexVAR model (Jiao et al., 2025), we follow the officially specified sampling parameters (cfg = 2.5, top_p = 0.95, top_k = 900, without using more-smooth) to collect and sample the LSRS training data. Additionally, since each scale in FlexVAR is merely a downsampled version of the original feature map, the scoring model in LSRS for FlexVAR receives $r_k$ instead of $e_k$. All other settings of LSRS remain consistent with those in VAR.

# B   MORE EXPERIMENTS ON VAR

Table 4: **More experiments on VAR**. Generative model comparison on class-conditional ImageNet (Deng et al., 2009) 256×256. Metrics include Fréchet inception distance (FID), inception score (IS), precision (Pre) and recall (rec). Step: the number of model runs needed to generate an image. Time: the relative inference time of VAR-$d$30. LSRS is applied to VAR models at various depths, achieving improvements across all cases.

| Model | FID↓ | IS↑ | Pre↑ | Rec↑ | Param | Step | Time |
|---|---|---|---|---|---|---|---|
| VAR-$d$16 (Tian et al., 2024) | 3.36 | 274.5 | 0.84 | 0.51 | 310M | 10 | 0.20 |
| **+ LSRS** $M = 4$ | 3.19 | **278.1** | 0.82 | 0.54 | 310M+4M | 10 | **0.21** |
| **+ LSRS** $M = 128$ | **2.97** | 276.4 | 0.81 | 0.55 | 310M+4M | 10 | 0.30 |
| VAR-$d$20 | 2.70 | 302.9 | 0.83 | 0.56 | 600M | 10 | 0.30 |
| **+ LSRS** $M = 4$ | 2.59 | **304.8** | 0.81 | 0.59 | 600M+4M | 10 | **0.31** |
| **+ LSRS** $M = 128$ | **2.54** | 303.9 | 0.81 | 0.59 | 600M+4M | 10 | 0.41 |
| VAR-$d$24 | 2.15 | 311.6 | 0.82 | 0.59 | 1.0B | 10 | 0.50 |
| **+ LSRS** $M = 4$ | 2.09 | **313.2** | 0.82 | 0.60 | 1.0B+4M | 10 | **0.51** |
| **+ LSRS** $M = 128$ | **2.03** | 312.6 | 0.82 | 0.60 | 1.0B+4M | 10 | 0.62 |

Table 4 shows that LSRS consistently improves performance across VAR models of different depths, demonstrating the stability of LSRS.

## C  CONSUMPTION OF LSRS

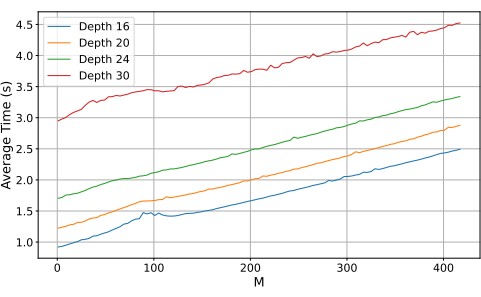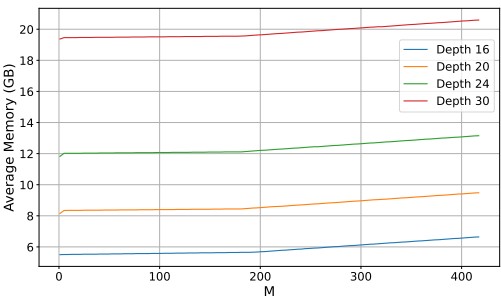

Figure 6: The left and right figures show the average time consumption and GPU memory usage of VAR with LSRS when generating 24 images in parallel using classifier-free guidance (CFG) with increasing $M$. The results are obtained by averaging over 3 runs.

## D  MORE SCORE DISTRIBUTION

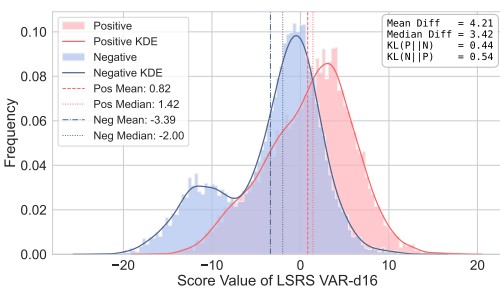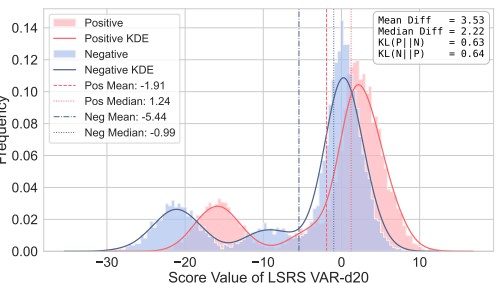

Figure 7: Left: VAR-$d16$ score distribution. Right: VAR-$d20$ score distribution.

## E  DETAILED DATA OF HYPERPARAMETER ABLATION

Table 5: Metrics across $M$ values with $ST = 2$

| $M$ | FID↓ | IS↑ | sFID↓ |
|---|---|---|---|
| – | 1.95 | 303.1 | 8.50 |
| 2 | 1.87 | 304.6 | 7.62 |
| **4** | 1.78 | **305.9** | 7.11 |
| 8 | 1.73 | 303.4 | 6.73 |
| 16 | 1.71 | 302.2 | 6.60 |
| 32 | 1.68 | 300.8 | 6.38 |
| 64 | 1.67 | 301.1 | 6.40 |
| **128** | **1.66** | 298.9 | 6.41 |
| 256 | 1.70 | 297.1 | 6.44 |
| 512 | 1.78 | 295.6 | 6.48 |
| 1024 | 1.75 | 296.3 | 6.57 |

Table 6: Metrics across $ST$ values with $M = 32$

| $ST$ | FID↓ | IS↑ | sFID↓ |
|---|---|---|---|
| – | 1.95 | 303.1 | 8.50 |
| 1 | 2.30 | 295.0 | 8.82 |
| **2** | **1.68** | 300.8 | 6.38 |
| 3 | 1.79 | 301.3 | 7.18 |
| 4 | 1.82 | 301.9 | 7.51 |
| 5 | 1.90 | 301.1 | 7.82 |
| 6 | 1.91 | 301.5 | 7.99 |
| 7 | 1.92 | 302.5 | 8.25 |
| 8 | 1.93 | 303.2 | 8.42 |
| 9 | 1.95 | 302.0 | 8.46 |
| **10** | 1.95 | **303.3** | 8.52 |

# F  ADDITIONAL ANALYSIS OF $M$

In this section, we conduct a more in-depth analysis of the impact of $M$ on LSRS. The Fréchet Inception Distance (FID) (Heusel et al., 2017) is decomposed as:

$$\text{FID} = \|\mu_r - \mu_g\|^2 + \text{Tr}(\Sigma_r) + \text{Tr}(\Sigma_g) - 2\,\text{Tr}\left((\Sigma_r\Sigma_g)^{1/2}\right),$$

We denote $\text{Tr}(\Sigma_r) + \text{Tr}(\Sigma_g) - 2\,\text{Tr}\left((\Sigma_r\Sigma_g)^{1/2}\right)$ as trace_term, and $\|\mu_r - \mu_g\|^2$ as mean_diff$^2$.

Table 7: Variation of FID and its components with $M$. For all $M$, $\text{Tr}(\Sigma_r) = 180.609$.

| $M$ | FID | mean_diff² | trace_term | $\text{Tr}(\Sigma_g)$ | $2\,\text{Tr}\left((\Sigma_r\Sigma_g)^{1/2}\right)$ |
|---|---|---|---|---|---|
| 1 | 1.954 | 0.229 | 1.725 | 172.276 | 351.160 |
| 2 | 1.869 | 0.227 | 1.642 | 172.600 | 351.566 |
| 4 | 1.781 | 0.227 | 1.554 | 172.855 | 351.910 |
| 8 | 1.729 | 0.216 | 1.513 | 173.407 | 352.502 |
| 16 | 1.707 | 0.217 | 1.490 | 173.331 | 352.449 |
| 32 | 1.683 | 0.210 | 1.473 | 173.847 | 352.983 |
| 64 | 1.676 | 0.203 | 1.473 | 174.232 | 353.368 |
| 128 | 1.663 | 0.201 | 1.462 | 174.518 | 353.664 |
| 256 | 1.702 | 0.206 | 1.496 | 174.797 | 353.909 |
| 512 | 1.775 | 0.208 | 1.567 | 174.794 | 353.836 |

In Table 7, we collected statistics on VAR-$d30$ with fixed $ST = 2$ and varying values of $M$, recording the FID and its individual components. The term mean_diff$^2$ measures the squared Euclidean distance between the mean feature vectors of generated and real images. Lower values indicate better alignment. Its trend mirrors that of FID: it steadily decreases from $M = 1$ to a minimum at $M = 128$, then slightly rebounds. This decreasing trend indicates that LSRS effectively corrects the model's systematic bias. As $M$ increases, the average quality or "correctness" of generated images improves continuously, peaking at $M = 128$.

The term trace_term reflects the discrepancy between the covariance matrices of the two distributions, primarily capturing differences in the "shape" or "diversity" of the distributions. A smaller trace_term indicates that the covariance structure of the generated samples more closely matches that of the real samples. Its trend also aligns with FID: decreasing from $M = 1$ to $M = 128$, then increasing for $M > 128$. This suggests that for $M \leq 128$, LSRS successfully brings the distribution of generated samples closer to the real data distribution. However, when $M > 128$, the rejection sampling becomes too aggressive, causing the model to heavily rely on a few high-scoring modes. This undermines the covariance structure of the generated feature distribution, leading the overall generated set to deviate macroscopically from the real data distribution.

Regarding the slight increase in mean_diff$^2$ when $M > 128$, we analyze it from the perspective of the overall distribution: at this stage, the model excessively concentrates on generating images from only a few high-scoring modes, essentially a subset of the real data. Consequently, the statistical center (mean) of the generated sample set is likely to shift away from that of the real data, resulting in a slight degradation in mean alignment.

Therefore, the conclusion is that when $M < 128$, LSRS gradually corrects the model's systematic bias, aligning the distribution of generated data with that of the real data. When $M > 128$, the model becomes overly focused on generating a few high-scoring modes, causing the mean and distribution structure of the generated data to deviate from those of the real data.

# G  GREEDY SELECTION VS. TOP-$k$ SAMPLING

In LSRS, the Best-of-N strategy greedily selects the candidate token map with the highest score among all options. A reasonable extension would be to use top-$k$ instead, which might lead to better results. So we applied top-$k$ sampling in this section instead of greedy selection in LSRS at each scale, which means sampling from the $k$ highest-scoring tokens according to a softmax of their scores.

Table 8: Performance of top-$k$ sampling in LSRS with $M = 128$ (VAR-$d30$, $ST = 2$)

| $k$ | FID | IS |
| --- | --- | --- |
| 1 | 1.66 | 298.9 |
| 2 | 1.63 | 299.0 |
| 4 | 1.66 | 297.5 |
| 8 | 1.65 | 297.4 |
| 16 | 1.68 | 301.0 |

Table 9: Performance of top-$k$ sampling in LSRS with $M = 256$ (VAR-$d30$, $ST = 2$)

| $k$ | FID | IS |
| --- | --- | --- |
| 1 | 1.70 | 297.1 |
| 2 | 1.71 | 299.4 |
| 4 | 1.68 | 299.4 |
| 8 | 1.71 | 296.8 |
| 16 | 1.70 | 298.1 |

In Table 8 and Table 9, we observe that top-$k$ sampling performs slightly better than greedy sampling ($k = 1$) in some cases, but the improvement is marginal and unstable. Moreover, for the same $k$, $M = 128$ consistently outperforms $M = 256$. This indicates that the degradation of LSRS when $M > 128$ is not caused by the greedy strategy.

# H   SCORING MODEL WITHOUT THE FIRST SCALE

As shown in Figure 5, after training, the model's accuracy on the first scale is only 52.2%, which is close to random guessing. This indicates that the data from the first scale indeed acts almost entirely as noise during training, making it very difficult for the model to learn meaningful distinctions. Possible reasons include:

- The first scale contains only a single token, lacking explicit structural information;
- The VAR generation at the first scale already nearly overlaps with the true data distribution.

Given this, excluding the first scale during scoring model training might reduce noise interference and lower the learning difficulty, potentially leading to better performance. However, from a theoretical standpoint, since the scoring model takes the scale index $k$ as a conditional input, the difficulty in learning the first scale should not heavily affect the learning of other scales.

Table 10: Performance comparison w and w/o the first scale

| scale | w first scale | w/o first scale | change |
|-------|---------------|-----------------|--------|
| 1     | 52.3%         | —               | —      |
| 2     | 76.3%         | 76.2%           | -0.1%  |
| 3     | 78.6%         | 78.5%           | -0.1%  |
| 4     | 77.2%         | 77.2%           | 0.0%   |
| 5     | 79.9%         | 79.9%           | 0.0%   |
| 6     | 82.4%         | 82.3%           | -0.1%  |
| 7     | 79.7%         | 79.8%           | +0.1%  |
| 8     | 91.0%         | 90.9%           | -0.1%  |
| 9     | 91.8%         | 91.7%           | -0.1%  |
| 10    | 84.3%         | 84.4%           | +0.1%  |

Table 10 shows the accuracy of the trained scoring models on each scale. As can be observed, there is virtually no significant difference in performance across scales when excluding the first scale. This suggests that, although the first scale contributes little signal, its inclusion does not meaningfully harm the model's ability to learn on the remaining scales.

# I   MORE SAMPLES

In this section, we present additional generated images from VAR and LSRS. Figure 8 illustrates the impact of parameter $M$ on the results in LSRS. Figure 9 demonstrates the effect of parameter $ST$ on the results in LSRS.

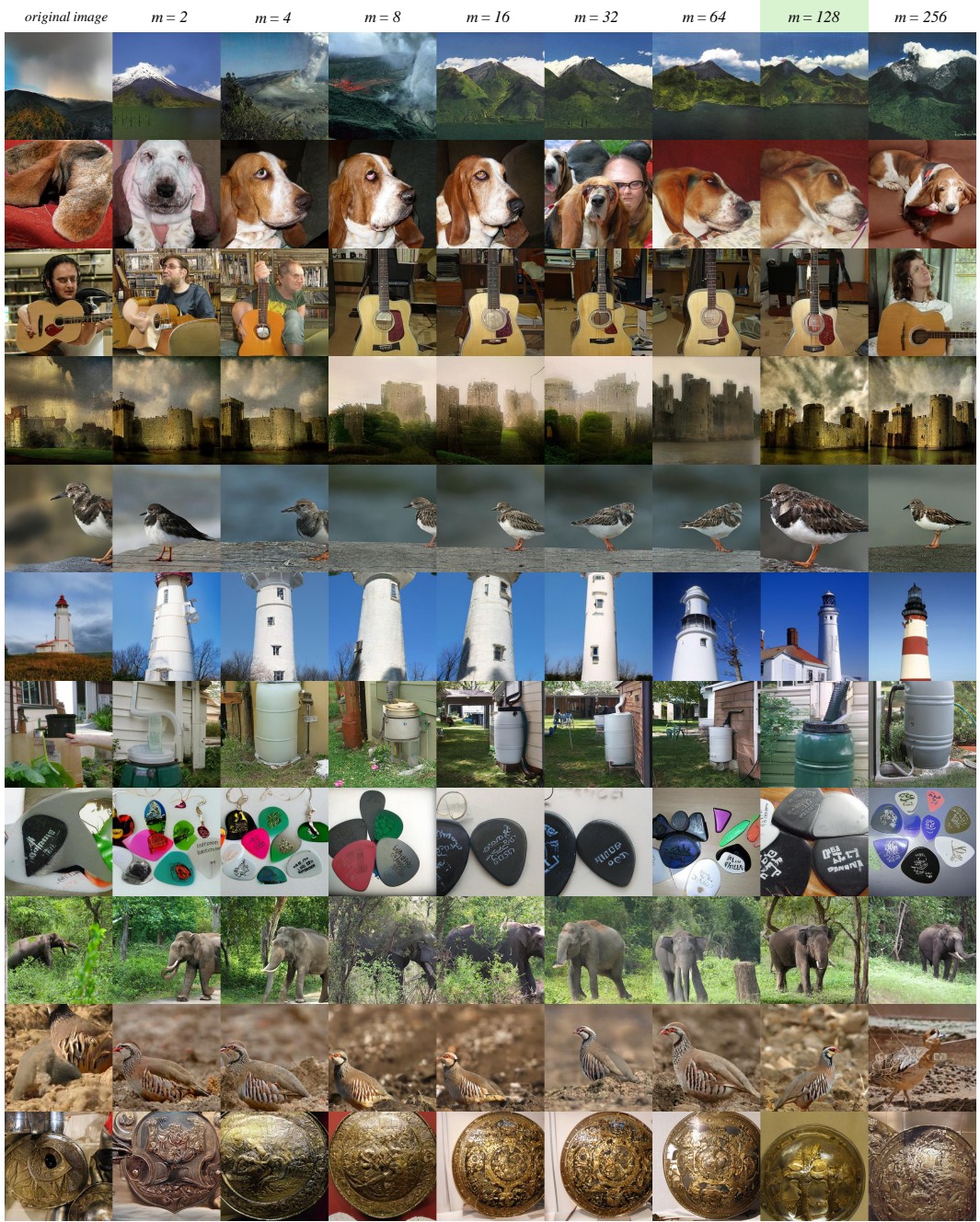

Figure 8: **Additional generated image demonstrations.** The leftmost column shows the original images generated by VAR-$d30$. The remaining columns display images generated by VAR-$d30$ + LSRS with fixed $ST = 2$, where only $M$ varies from left to right.

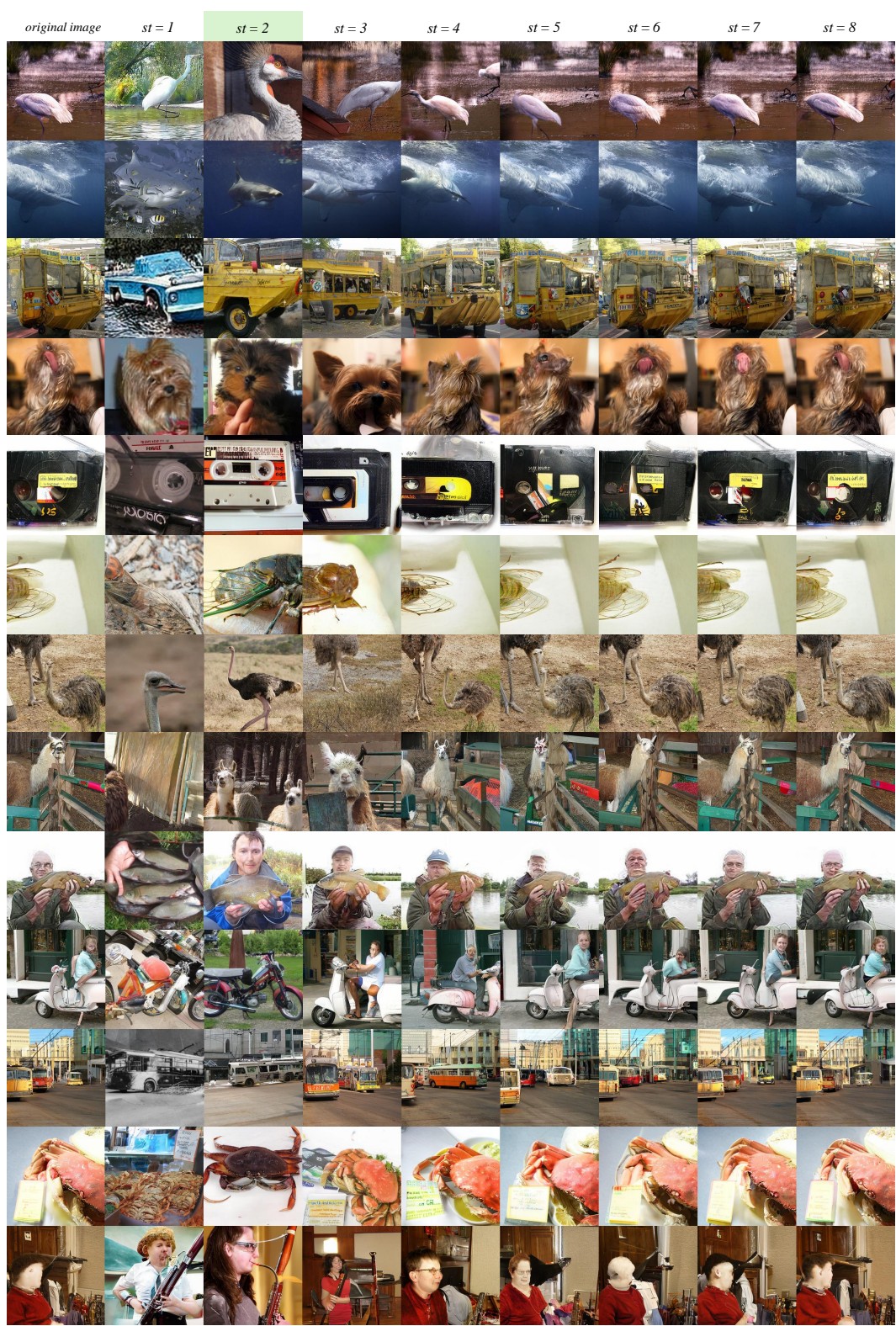

Figure 9: **Additional generated image demonstrations.** The leftmost column shows the original images generated by VAR-$d$30. The remaining columns display images generated by VAR-$d$30 + LSRS with fixed $M = 64$, where only $ST$ varies from left to right.

## J    THE USE OF LARGE LANGUAGE MODELS

In the preparation of this manuscript, a Large Language Model (LLM) was used solely for the purpose of language polishing and stylistic refinement of the text. The LLM was prompted to improve clarity, grammar, and fluency of expression, without altering the core scientific content, methodology, results, or interpretations presented in the paper. The research ideas, experimental design, data analysis, and original writing were entirely conducted by the human authors. The LLM did not contribute to the generation of hypotheses, formulation of research questions, or development of novel concepts. Its role was strictly limited to post-writing linguistic enhancement.

