# OpenReview forum: "LSRS: Latent Scale Rejection Sampling for Visual Autoregressive Modeling"
_ICLR.cc/2026/Conference — Submitted to ICLR 2026_

### Official Review · Reviewer_kcQM · 2025-10-28

**Soundness:** 2
**Presentation:** 3
**Contribution:** 2
**Rating:** 4
**Confidence:** 5

**Summary:**

This paper proposes Latent Scale Rejection Sampling (LSRS), a test time scaling method for visual autoregressive (VAR) models. LSRS introduces lightweight scoring and rejection sampling at each layer during inference, prioritizing early scales that most affect structural quality. Experimental results demonstrate that LSRS significantly improves generation quality (e.g., FID reduced from 1.95 to 1.66).

**Strengths:**

- The exploration of TTS for VAR is a previously unexplored area, and this work provides valuable insights for future research.

- The proposed method achieves further performance improvements over the original VAR through test-time computation.

- The paper is well-written and easy to follow.

**Weaknesses:**

- Equation 2 might be incorrect. In VAR generation, $r_k(i,j)$ should also be conditioned on other tokens at the same scale, but this is not included in the equation.

- The proposed idea seems somewhat too simple: training a classifier to assess the quality of token maps across different scales.

- The effectiveness of the proposed TTS approach for VAR does not show computation scaling property, as shown in Figure 3. Although I appreciate the authors’ detailed explanation of this phenomenon, it still affects the practicality of the proposed method.

- The paper mainly uses BoN and top-k methods for TTS. I wonder whether more advanced approaches, such as beam search, could further improve performance.

**Questions:**

- The proposed method is mainly validated on class-conditioned image generation tasks. How does it perform on text-to-image T2I tasks, such as using the VAR-based T2I model Infinity[1]?

- The proposed method scales test time to enhance the generation quality of the VAR model. It would thus be interesting to apply the proposed TTS method to the VAR-acceleration approaches such as FastVAR [2], to see whether the resulted variants could achieve comparable efficiency to the baseline model but delivering superior generation results?

- I would be glad to increase my rating if the authors could address the above concerns.

> [1] Infinity : Scaling Bitwise AutoRegressive Modeling for High-Resolution Image Synthesis. CVPR25

> [2]FastVAR: Linear Visual Autoregressive Modeling via Cached Token Pruning. ICCV25

---

> ### Author Response · Authors · 2025-11-17
>
> Thank you for recognizing that our paper is well-written and provides new insights. We have responded to your questions and conducted some additional experiments. We hope this will help you better evaluate our work:
>
> ---
>
> > Equation 2 might be incorrect. In VAR generation, $r_k(i,j)$ should also be conditioned on other tokens at the same scale, but this is not included in the equation.
>
> Your statement may contain some ambiguity. Let me clarify it more precisely:
>
> * Equation 2 **correctly** describes the sampling process of the VAR model. This is because, within a single scale of VAR, the probability distributions of all tokens are computed in parallel, and they are sampled in parallel as well. That is, the sampling result of $r_k(i,j)$ does not affect the distribution of $r_k(i',j')$ (where $(i,j) \neq (i',j')$ but they belong to the same scale). This means all tokens within a scale are sampled independently, exactly as described by Equation 2.
>
> * However, Equation 2 **incorrectly** models the probability distribution of $r_k$. As you pointed out, once a particular $r_k(i,j)$ is sampled, its sampling result actually affects the distributions of other tokens within the same scale. This is precisely the issue discussed in lines 160–180 of the paper.
>
> * Therefore, from the above two points we can conclude that the sampling process of VAR is **flawed**. Thus, we propose LSRS to implicitly model the dependencies among tokens within the same scale, thereby improving generation quality.
>
> > The paper mainly uses BoN and top-k methods for TTS. I wonder whether more advanced approaches, such as beam search, could further improve performance.
>
> This is possible, but it seems to deviate from the main focus of our work. In addition, beam search introduces substantial computational overhead, which goes against the design goal of LSRS, namely efficiency. However, to verify whether it could improve LSRS, we conducted related experiments, and the results are as follows:
>
> |K|FID|IS|Time|
> |--|----|---|----|
> |-|1.95|303.1|1.00|
> |1|1.68|300.8|1.04|
> |2|1.66|399.5|>2|
> |4|1.65|299.1|>4|
>
> All experiments were conducted on VAR-$d30$, with LSRS parameters $ST=2, M=32$. When $K=1$, beam search degenerates into BoN. As we can see, beam search has very limited impact on the metrics, yet it significantly increases computational cost. Other, more complex sampling strategies are unlikely to work either. Therefore, in practical applications, **simply using the most basic and stable BoN is already sufficient**.
>
> > The proposed method is mainly validated on class-conditioned image generation tasks. How does it perform on text-to-image T2I tasks, such as using the VAR-based T2I model Infinity[1]?
>
> Since the training data of Infinity is not publicly available, we are currently unable to apply LSRS to Infinity. In addition, for T2I models, the LSRS scoring model needs to be modified. For example, replacing class conditioning with text conditioning requires adding a text encoder. This is an extension of LSRS and would be very interesting, but it also implies a large amount of additional work and exploration. We believe this is better suited for a separate follow-up work dedicated to LSRS.

---

> ### Author Response · Authors · 2025-11-17
>
> > The proposed method scales test time to enhance the generation quality of the VAR model. It would thus be interesting to apply the proposed TTS method to the VAR-acceleration approaches such as FastVAR [2], to see whether the resulted variants could achieve comparable efficiency to the baseline model but delivering superior generation results?
>
> The FastVAR paper includes experiments in its appendix that combine their method with the original VAR. They found that FastVAR performs **poorly** on the original VAR. Since they only provide implementations of FastVAR for HART and Infinity, we re-implemented FastVAR for VAR based on their code and conducted experiments combining it with LSRS. The results are as follows:
>
> |Model|FID|IS|Time|
> |-----|----|---|----|
> |VAR-$d30$|1.95|303.1|1.00|
> |VAR-$d30$ + FastVAR (13:0.50, 16:0.75)|3.83|239.4|0.81|
> |VAR-$d30$ + FastVAR (13:0.25, 16:0.50)|3.17|254.6|0.90|
> |VAR-$d30$ + FastVAR (13:0.00, 16:0.25)|2.14|288.3|1.07|
> |VAR-$d30$ + LSRS|1.68|300.8|1.04|
> |VAR-$d30$ + LSRS + FastVAR (13:0.50, 16:0.75)|3.68|237.8|0.83|
> |VAR-$d30$ + LSRS + FastVAR (13:0.25, 16:0.50)|2.95|252.5|0.92|
> |VAR-$d30$ + LSRS + FastVAR (13:0.00, 16:0.25)|1.95|285.4|1.10|
>
> The LSRS parameters are $ST=2, M=32$. The parameters of FastVAR are expressed as (scale size : pruning ratio). For example, (13:0.50, 16:0.75) means pruning 50% of the tokens at the scale of size 13 and pruning 75% of the tokens at the scale of size 16, while no pruning or acceleration is applied to the other scales.
>
>
> The results above differ from those reported in the FastVAR paper because this is our own implementation. Moreover, they do not seem to report what hyperparameters were used for FastVAR on VAR, so we could only try several ourselves.
>
> The results show that although FastVAR provides some speedup, it significantly degrades the generation quality of VAR. When using FastVAR with parameters (13:0.00, 16:0.25), it is even slower than not using it. One possible reason is that FastVAR is mainly designed for high-resolution image generation, and the paper validates its performance on two T2I models (HART and Infinity). Compared with VAR, these two models have more scales and larger scale sizes. Specifically:
>
> * HART (1024×1024): 1, 2, 3, 4, 5, 7, 9, 12, 16, 21, 27, 36, 48, 64
> * Infinity (1024×1024): 1, 2, 4, 6, 8, 12, 16, 20, 24, 32, 40, 48, 64
> * VAR (256×256): 1, 2, 3, 4, 5, 6, 8, 10, 13, 16
>
> Therefore, we speculate that FastVAR may not be suitable for low-resolution original VAR models. For high-resolution HART and Infinity, the later large scales contain information after many residual steps: they have many tokens, but each individual token has relatively small impact on the final result. Thus, FastVAR can maintain relatively good quality while accelerating. In contrast, for the original VAR, each token carries more information than in HART or Infinity. As a result, FastVAR causes more information loss, leading to significant quality degradation. Moreover, VAR (or VAR+LSRS) is already sufficiently fast, and if the hyperparameters are not properly tuned, FastVAR may even introduce additional overhead.
>
> Combining other VAR-acceleration approaches with LSRS may also not be a simple additive process. They may interact with each other, requiring careful tuning and exploration. This is likely more suitable as a separate follow-up research direction.

---

> ### Author Response · Authors · 2025-11-25
>
> Dear Reviewer kcQM
>
> Thank you very much for your detailed review comments. We have addressed your concerns in our rebuttal and included additional experiments. We hope you can find some time to review our responses. Should you have any further questions, please feel free to let us know.
>
> Best
>
> Authors

---

> > ### Comment · Reviewer_kcQM · 2025-11-25
> > **Response to Author Rebuttal**
> >
> > Thanks for your reply.
> > After reading the rebuttal, I am still confusing with the formulation in Eq.2 of the paper. As the attention mask in Fig.4 of the original VAR [A] paper shows, the r_k(i,j) can actually attened its same-scale r_k(i', j'). However, the authors state that these two elements are modeled independently, which is incorrect from my view. Is there any part that I misunderstand?
> >
> > > [A] Visual Autoregressive Modeling: Scalable Image Generation via Next-Scale Prediction

---

> > > ### Author Response · Authors · 2025-11-25
> > >
> > > We appreciate the reviewer’s careful examination of the attention mechanism in VAR. We agree that the attention mask in VAR allows for interaction of same-scale tokens, but we would like to clarify the subtle but crucial distinction between **feature-level interaction** (via attention) and **probabilistic independence** (during sampling). The explanation below will definitely help:
> > >
> > > ### Feature Interaction vs. Sampling Independence
> > >
> > > * **Interaction among same-scale tokens**: Your statement, “$r_k(i,j)$ can actually attend to $r_k(i',j')$ at the same scale,” is **entirely correct**. But we must emphasize that this interaction exists **only at the feature level**. In each Transformer layer, tokens at the same scale can exchange information with each other.
> > >
> > > * **Computing each token’s probability distribution within the same scale**: After the Transformer forward pass finishes, each token simultaneously and in parallel passes through a linear layer (classification head) to produce its probability distribution. At this moment, these distributions are already **fixed**. Specifically, we now have the probability distributions for $r_k(i,j)$ and $r_k(i',j')$, but their actual sampled values (token IDs) have not yet been drawn.
> > >
> > > * **Sampling tokens at the same scale**: During sampling, the final values of $r_k(i,j)$ and $r_k(i',j')$ are drawn **simultaneously**. That is, the sampling result of $r_k(i,j)$ (e.g., ID = 42) does **not** affect the distribution of $r_k(i',j')$, nor its sampled result. This is exactly what we mean by **independence** of same-scale tokens.
> > >
> > > ### Mathematical Proof via Chain Rule
> > >
> > > After understanding the above, let’s prove Eq. 2 in the paper—it will be very quick. Denote the tokens inside $r_k$ as $z_1, z_2, \ldots, z_N$ (where $N = h_k \times w_k$). According to the probability chain rule, the joint probability can always be decomposed as:
> > >
> > > $$
> > > p(r_k \mid r_{<k})
> > > = p(z_1 \mid r_{<k}) \cdot p(z_2 \mid z_1, r_{<k})
> > > \cdots p(z_N \mid z_1, \ldots, z_{N-1}, r_{<k})
> > > $$
> > >
> > > **Here is the key!:** In a typical autoregressive scenario (such as raster-scan generation), the distribution of $z_m$ indeed depends on the **sampled values** of previous tokens $z_{1:m-1}$. However, under VAR’s parallel decoding mechanism, the distribution of each $z_m$ is computed solely from $r_{<k}$. When computing the distribution of $z_2$, the model has **no access** to the sampled value of $z_1$. Therefore, the **conditional dependency disappears**:
> > >
> > > $$
> > > p(z_m \mid z_1, \ldots, z_{m-1}, r_{<k})
> > > = p(z_m \mid r_{<k})
> > > $$
> > >
> > > Plugging this into the chain rule immediately gives Eq. 2:
> > >
> > > $$
> > > p(r_k \mid r_1, r_2, \ldots, r_{k-1})
> > > = \prod_{i=1}^{h_k} \prod_{j=1}^{w_k}
> > > p(r_k(i,j) \mid r_1, r_2, \ldots, r_{k-1})
> > > $$
> > >
> > > This is precisely the mathematical statement of the **independence** we discussed.
> > >
> > > ### The "Rock–Paper–Scissors" Analogy
> > >
> > > If you still find this confusing, the following “Rock–Paper–Scissors” analogy should make everything clear:
> > >
> > > * Attention phase (***your interaction occurs here***): Before the game starts, both players observe each other, communicate, and pay attention to each other's body language. There is information exchange during this process.
> > > * Sampling phase (**parallel generation**): Both players reveal their gesture simultaneously.
> > > * **Independence**: At the moment the gestures are revealed, Player A’s choice cannot be conditioned on Player B’s choice, and vice versa. Despite prior interaction, the final outputs are independently sampled. The joint probability is the product of the marginals: $P(A, B) = P(A) \cdot P(B)$.
> > >
> > > Thus, there is no contradiction. The interaction occurs at the feature level, while the independence described in Eq. 2 governs the sampling process. Ideally, tokens should depend on each other's sampled values to ensure structural consistency, but they cannot do so due to parallel decoding. This is precisely the limitation our paper identifies and solves with LSRS.

---

> ### Author Response · Authors · 2025-11-27
>
> Dear Reviewer kcQM
>
> Thank you very much for your thoughtful consideration and careful review of the VAR model and our proposed LSRS. We have provided a more detailed response to your questions, which we believe will help you fully understand our approach. Should you have any further questions, please feel free to raise them at any time.
>
> Best
>
> Authors

---

### Official Review · Reviewer_qV4B · 2025-10-29

**Soundness:** 3
**Presentation:** 3
**Contribution:** 3
**Rating:** 4
**Confidence:** 4

**Summary:**

This paper introduces LSRS, a method that progressively refines token maps in the latent scale during inference to enhance VAR models. It first analyzes the structural errors observed in images generated by VAR models and attributes them to the imperfect parallel sampling mechanism, particularly at the early scales. Motivated by this finding, LSRS performs latent-scale rejection sampling, where a scoring model is trained to select the intermediate token maps with the highest predicted quality score. Experimental results demonstrate that LSRS improves the image generation quality of VAR models while incurring only minimal additional computational overhead.

**Strengths:**

- The analysis of VAR’s mechanisms and inherent limitations provides valuable insights, revealing that earlier scales play a more critical role in determining overall image structure.
- The proposed latent scale rejection sampling (LSRS) method is technically well-founded, combining real and synthetic data construction, scoring model training, and token map selection guided by the scoring model.
- Extensive experiments on the ImageNet image generation task demonstrate the effectiveness of LSRS across various VAR base models, while comprehensive ablation studies offer further insights into its design and performance.

**Weaknesses:**

- The relationship between the imperfect parallel sampling mechanism and the proposed LSRS sampling method is not clearly explained. Since the base VAR models remain unchanged and LSRS just runs the base models several times for certain latent scales, it is unclear how LSRS effectively mitigates the limitations of the mutually independent token sampling mechanism.
- While the ImageNet experiments are sufficient to demonstrate the effectiveness of the proposed method, the paper would be further strengthened by including results on text-to-image (T2I) tasks and evaluating additional metrics. This could be achieved by extending experiments to VAR variants designed for T2I generation, such as HART [1] and Infinity [2].
- Some of the experimental analyses are not entirely convincing:
  - Line 370 states that "Applying LSRS from the first scale causes many samples to converge to similar values at this scale, thereby reducing generation diversity and degrading FID." However, as shown in Figure 3, the IS score also decreases in this case. In my understanding, a lower diversity would not reduce IS.
  - Similarly, line 411 claims that "This may again be attributed to the fact that excessively large M could reduce the diversity of generated images." Yet, IS still drops as M increases, which seems inconsistent with this explanation.
- There are a few typographical errors in the manuscript that should be carefully corrected. For example:
  - Line 196, "Brock et al., **1809**"

[1] HART: Efficient Visual Generation with Hybrid Autoregressive Transformer
[2] Infinity: Scaling Bitwise AutoRegressive Modeling for High-Resolution Image Synthesis

**Questions:**

- My understanding of the experimental setup is as follows: there are K scales in VAR image generation, where scale 1 contains a single token and scale K has the largest token map. From scale 1 to scale ST−1, only one sample is generated, while from scale ST to scale K, M samples are drawn. If this understanding is correct, several questions arise:
  - The previous analysis states that "Therefore, if we can optimize th early scale, we can then efficiently improve the quality of the final images." This seems inconsistent with the implementation, where multiple samples are drawn only from scale ST to scale K, rather than from the early scales that are claimed to be more critical.
  - It is also stated that LSRS introduces only minimal computational overhead. However, under the ST=2, M=4 setting, all scales except scale 1 appear to be sampled four times, which would result in nearly 4x computational cost. The reported overhead of only 0.01x is therefore unclear and warrants further explanation.
- In line 265, it is stated that "To prevent data leakage, the random seeds used for constructing the VAR sampling dataset are different from those employed during evaluation". Could the authors elaborate on how random seeds are used in this process and how their reuse could cause data leakage?
- Regarding the pairwise loss defined in Equation (4), could the authors clarify how the pairs are formed between real samples and synthetic samples during training?

---

> ### Author Response · Authors · 2025-11-15
>
> Thank you for recognizing the novel insights in our paper and the thoroughness of our experiments. However, we feel that there may be some significant misunderstandings regarding the core of our work. We hope our response can help clarify these points and assist you in better evaluating our contribution:
>
> ---
>
> >  The relationship between the imperfect parallel sampling mechanism and the proposed LSRS sampling method is not clearly explained. Since the base VAR models remain unchanged and LSRS just runs the base models several times for certain latent scales...
> >
> > It is also stated that LSRS introduces only minimal computational overhead. However, under the ST=2, M=4 setting, all scales except scale 1 appear to be sampled four times, which would result in nearly 4x computational cost. The reported overhead of only 0.01x is therefore unclear and warrants further explanation.
>
> It seems there is a major misunderstanding here: **LSRS does not run the VAR model multiple times; it only samples multiple times from a single computed distribution**. At each scale $k$, the VAR model performs **only one** forward pass, which is used to compute the probability distribution of all tokens at that scale, $p(r_k \mid r_{<k})$. The key steps of LSRS happen **afterward**:
>
> * **Sampling (this is likely where your misunderstanding occurs)**: We draw $m_k$ candidate token maps ${r_k^{(i)}}$ from this (imperfect) distribution $p(r_k \mid r_{<k})$. **This process is fully parallelized and introduces almost no additional cost compared to sampling once in the original VAR model.**
>
> * **Scoring**: We use a lightweight scoring model $S$ to evaluate each of the $m_k$ candidate maps. In fact, this is where most of LSRS’s overhead resides. But since we batch all sampled candidates together for parallel evaluation, and since the scoring model is very small, the overall cost remains low.
>
> * **Selection**: We pick the candidate map with the highest score as the final output for scale $k$. This step introduces virtually no additional overhead.
>
> All of the above is clearly and rigorously described in Algorithm 1 of the paper.
>
> > It is unclear how LSRS effectively mitigates the limitations of the mutually independent token sampling mechanism.
>
> LSRS adopts a **“propose–evaluate–select”** strategy, and it does not directly alter the forward process of the VAR model. The scoring model is trained to **implicitly capture the dependencies among all tokens within a sample $r_k$**. When LSRS selects the highest-scoring map from the $m_k$ candidates, it is effectively choosing the sample that best matches the true internal token structure (i.e., the one with the highest implicit joint probability). This step effectively filters out low-quality token maps such as those with structural errors that arise from deficiencies in VAR’s independent sampling.
>
> To illustrate this with an **extreme example**:
>
> Suppose we use a **purely random** proposer. For instance, sampling tokens uniformly from the vocabulary to construct $r_k$. Without an evaluator, this would only produce noise. However, if we also had a **perfect** scoring model, which could evaluate the realism of any token map $r_k$ with 100% accuracy. Then, in theory, we would only need to sample enough candidates ($m_k$ sufficiently large) from this random proposer, and the perfect evaluator would be able to pick out a meaningful sample with the correct spatial structure.
>
> Now, in **LSRS**:
>
> * We do not use a “random proposer,” because that would be far too inefficient. Instead, we use VAR as a proposer that is imperfect but sufficiently strong. The samples generated by VAR are far better than random noise, yet they can still occasionally be low-quality.
> * We do not rely on a “perfect evaluator,” which of course is impossible to obtain. Instead, we train a lightweight but effective scoring model. The training objective of this scoring model is precisely to implicitly capture all token-level dependencies within a sample $r_k$, achieved by distinguishing between real (structurally correct) token maps and generated (potentially structurally flawed) ones.

---

> ### Author Response · Authors · 2025-11-15
>
> > The previous analysis states that "Therefore, if we can optimize th early scale, we can then efficiently improve the quality of the final images." This seems inconsistent with the implementation, where multiple samples are drawn only from scale ST to scale K, rather than from the early scales that are claimed to be more critical.
>
> **Our implementation is consistent with our analysis. In the main experiment (Table 1), $ST=2$, which is precisely the earliest scale that possesses spatial structure.** We highlight the importance of scales 2, 3, and 4 in line 186 of the paper and in Figure 1. These all belong to the early scales. Moreover, the ablation study in Table 3 demonstrates that the earlier LSRS is applied (with $ST \geq 2$, i.e., the closer to 2), the more significant the improvement in final image quality. Therefore, our implementation aligns well with our analysis.
>
> > Regarding the pairwise loss defined in Equation (4), could the authors clarify how the pairs are formed between real samples and synthetic samples during training?
>
> This is very intuitive: for any two data points, as long as one is a real sample and the other is a generated sample—while sharing the same class $c$ and scale $k$—they form a training pair. The training objective is to encourage the scoring model to assign higher scores to real samples than to generated ones, given the same class and scale.
>
> > In line 265, it is stated that "To prevent data leakage, the random seeds used for constructing the VAR sampling dataset are different from those employed during evaluation". Could the authors elaborate on how random seeds are used in this process and how their reuse could cause data leakage?
>
> Since training the scoring model requires using data generated by VAR, using the same random seed would cause the samples generated at the first scale where LSRS is applied during inference to potentially appear in the scoring model’s training set. This would constitute data leakage. To avoid this, the random seeds used for generating training data for the scoring model and those used during evaluation are kept different.

---

> ### Author Response · Authors · 2025-11-25
>
> Dear Reviewer qV4B
>
> Thank you very much for your detailed review comments. However, we noticed a significant misunderstanding regarding our proposed method, which we have clarified in our rebuttal. We have also addressed your other concerns. We hope you could kindly take a moment to review our responses when convenient. Should you have any further questions, please feel free to let us know at any time.
>
> Best
>
> Authors

---

> > ### Comment · Reviewer_qV4B · 2025-11-27
> >
> > I appreciate the authors for pointing out my misunderstanding. I understand the computational cost of LSRS now. However, several concerns remain:
> >
> > - As stated in my weakness session, several analysis attributes the degradation of FID to lower diversity. However, Inception Score also decreases in these settings. In my understanding, a lower diversity would not reduce IS.
> > - Typo
> >   - Line 196, "Brock et al., **1809**". The paper "Large scale gan training for high fidelity natural image synthesis" definitely was not published in the year 1809.
> > - For which scales to apply LSRS, the paper states that "Therefore, if we can optimize the early scale, we can then efficiently improve the quality of the final images." I'm wondering why LSRS is always applied from scale ST to the last scale K, rather than "early scales" (e.g., scale ST to scale ED, where ED < K).

---

> > > ### Author Response · Authors · 2025-11-27
> > >
> > > > For which scales to apply LSRS, the paper states that "Therefore, if we can optimize the early scale, we can then efficiently improve the quality of the final images." I'm wondering why LSRS is always applied from scale ST to the last scale K, rather than "early scales" (e.g., scale ST to scale ED, where ED < K).
> > >
> > > We sincerely apologize for having misunderstood your question in our previous response. Upon reflection, we recognize that your current query closely resembles Reviewer Wg8z’s Question 2. Therefore, the following reply may contain some repetition:
> > >
> > > We believe the experimental results presented in Figure 3 are already sufficient to demonstrate the importance of early-stage scales. However, we agree that adding an ablation study where LSRS is applied only on eraly scales would more strongly support our conclusion. Accordingly, **we have conducted this experiment, and the results are as follows**:
> > >
> > > | Scale  | FID (A)  | FID (B)   |
> > > |-----|-------|-------|
> > > | -   | 1.95  | 1.95 |
> > > | 1   | 2.30  | 2.49 |
> > > | 2   | 1.68  | 1.80 |
> > > | 3   | 1.79  | 1.88 |
> > > | 4   | 1.82  | 1.90 |
> > > | 5   | 1.90  | 1.93 |
> > > | 6   | 1.91  | 1.93 |
> > > | 7   | 1.92  | 1.94 |
> > > | 8   | 1.93  | 1.94 |
> > > | 9   | 1.95  | 1.95 |
> > > | 10  | 1.95  | 1.95 |
> > >
> > > The experiments are conducted on VAR-$d30$ with $M=32$ fixed. Here, FID (A) denotes applying LSRS starting from a given scale and continuing for all subsequent scales, while FID (B) denotes applying LSRS only at that specific scale and not at others. Regarding the $ED$ parameter you mentioned, we in fact implemented it exactly the same: each row in FID (B) corresponds to $ED = ST + 1$. We believe this setting is already sufficiently representative. From the results, we can see that using LSRS at early scales is indeed more beneficial. Moreover, FID (A) consistently outperforms FID (B), with a larger advantage at early scales. Therefore, **LSRS needs to be applied as early as possible and also maintained at later scales in order to achieve the best performance**.

---

> ### Author Response · Authors · 2025-11-27
>
> Thank you for your reply and for the careful review of our work. In response to the remaining concerns, we have conducted additional experiments and provided corresponding explanations. We hope this will help you better evaluate our research:
>
> ---
>
> > As stated in my weakness session, several analysis attributes the degradation of FID to lower diversity. However, Inception Score also decreases in these settings. In my understanding, a lower diversity would not reduce IS.
>
> Thanks to the reviewer for pointing out the relationship between diversity and the Inception Score. First, it is important to clarify that our paper only discusses class-agnostic diversity, namely **intra-class diversity**, which indeed does not directly affect the Inception Score. Therefore, to investigate the real cause of the IS drop, we decompose the IS metric for a more in-depth analysis:
>
> $$\ln(\text{IS}) = H(y) - \mathbb{E}[H(y|x)]$$
>
> Here, the marginal entropy $H(y)$ measures the class balance/diversity of generated images, i.e., whether the model generates different classes uniformly. This diversity refers to **inter-class diversity**, which should be distinguished from the **intra-class diversity** discussed in our paper. The conditional entropy $\mathbb{E}[H(y|x)]$ corresponds to the clarity/confidence of generated images, which refers to whether the images can be confidently classified into a specific category by a pretrained image classifier (Inception v3 in out paper).
>
> We compute the marginal entropy $H(y)$ and the conditional entropy $\mathbb{E}[H(y|x)]$ under different settings, and the results are shown in the table below (since we cached the activations for each setting during our prior evaluations, this was very fast):
>
>
> | Setting | FID | IS | ln(IS)  | Marginal Entropy | Conditional Entropy |
> |--------------------|--------|----------|--------|------------------|---------------------|
> | Original VAR       | 1.959  | 303.110  | 5.714  | 6.902            | 1.123               |
> | ST=1, M=32         | 2.305  | 294.684  | 5.686  | 6.901            | 1.150               |
> | ST=2, M=2          | 1.869  | 305.275  | 5.721  | 6.902            | 1.116               |
> | ST=2, M=4          | 1.781  | 305.910  | 5.723  | 6.903            | 1.115               |
> | ST=2, M=8          | 1.729  | 304.067  | 5.717  | 6.903            | 1.121               |
> | ST=2, M=16         | 1.707  | 301.393  | 5.708  | 6.903            | 1.127               |
> | ST=2, M=32         | 1.683  | 300.408  | 5.705  | 6.903            | 1.131               |
> | ST=2, M=64         | 1.676  | 300.902  | 5.707  | 6.903            | 1.131               |
> | ST=2, M=128        | 1.663  | 298.850  | 5.700  | 6.902            | 1.137               |
> | ST=2, M=256        | 1.702  | 296.889  | 5.693  | 6.903            | 1.143               |
> | ST=2, M=512        | 1.775  | 295.653  | 5.689  | 6.902            | 1.147               |
> | ST=2, M=1024       | 1.749  | 296.135  | 5.691  | 6.903            | 1.146               |
>
> From the above data, we observe that under all settings, the marginal entropy $H(y)$ remains stable (≈6.90). This indicates that the model never exhibits inter-class mode collapse; the class distribution consistently stays balanced. However, as $M$ increases, the conditional entropy $\mathbb{E}[H(y|x)]$ begins to rise slightly. This suggests that the model starts to favor generating safer samples with weaker class-specific characteristics, causing the score distribution produced by the classifier to become less sharp. The same interpretation also applies to the $ST=1, M=32$ configuration.
>
> Therefore, the decrease in Inception Score under certain LSRS settings is not due to reduced inter-class diversity, but rather because the **generated images become safer and more conservative**. In fact, this aligns with the conclusions drawn from our FID analysis as well.
>
> (Lastly, we would like to emphasize that the fluctuation magnitude of IS is consistently very small (≈1.4%). In contrast, the improvement in FID brought by LSRS is substantial (≈15%). Moreover, with properly chosen parameters (e.g., $ST=2, M=4$), both metrics can even improve simultaneously. Thus, we believe that our proposed LSRS is a genuinely effective and efficient method.)
>
>
> > Typo
> > Line 196, "Brock et al., 1809". The paper "Large scale gan training for high fidelity natural image synthesis" definitely was not published in the year 1809.
>
> Thank you for your careful review and for pointing out this typo. We have corrected it to “Brock et al., 2018,” and the revised PDF of the paper has been uploaded.

---

> ### Comment · Reviewer_qV4B · 2025-11-28
>
> Thanks the authors' reply. The analysis of inception score is insightful. Most of my concerns are addressed. I will raise my score.
>
> P.S. It seems the review and rating can not be edited now. I will raise the rating once it can be edited.

---

> > ### Author Response · Authors · 2025-11-28
> >
> > We are very glad that our response resolved your concerns! Once again, we sincerely appreciate your time and engagement in the discussion.

---

### Official Review · Reviewer_Wg8z · 2025-10-30

**Soundness:** 2
**Presentation:** 3
**Contribution:** 2
**Rating:** 4
**Confidence:** 4

**Summary:**

This paper proposes Latent Scale Rejection Sampling (LSRS) to mitigate the problem that parallel token sampling within a scale may lead to structural errors in VAR. The LSRS employs a lightweight scoring model to select the highest-quality one from generated multiple candidate token maps. Experiments demonstrate that LSRS improves VAR’s generation quality with minimal additional computational overhead.

**Strengths:**

1. The paper identifies a key limitation of current VAR models: parallel sampling many tokens within a scale in a single step may brings a degradation in the quality of generation.
2. The LSRS introduces the rejection sampling in the latent space of multi-scale autoregressive models, and it is simple and lightweight, making it highly practical for deployment.
3. The propose LSRS method improve the generation quality than VAR, which reduce its FID score from 1.95 to 1.78 while increasing the inference time by merely 1%.

**Weaknesses:**

1. The issue in this paper is that parallel token sampling within a scale may lead to structural errors, resulting in suboptimal generated images. To a certain extent, LSRS reduces the structural errors in the generation process, but it did not directly address the root cause of the problem, parallel sampling many tokens. Instead, It merely selected the best result from the candidate pool.
2. Figure 1 does not fully illustrate the issue that parallel token sampling within a scale may lead to structural errors. Because this structural error is caused by replacing the token maps, which merely indicates that errors at the early scales are earlier to affect the generation quality of the model.
3. Increasing the sampling steps in each scale is a more fundamental and direct method, which directly addresses the essence of the issue. However, there were no corresponding comparative experiments in the paper.
4. The ratio between computational cost and performance return is not adequately discussed in the later scales, because it is observed that earlier scales have a greater impact on the image structure in the paper.

**Questions:**

1. Why is it said that applying LSRS from the first scale causes many samples to converge to similar values at this scale in line 370? There is no direct evidence in the paper;
2. Why are no ablation experiments conducted on the later scales about the use or non-use of LSRS?
3. Can enhancing the ability of the score model improve the generation performance?

---

> ### Author Response · Authors · 2025-11-16
>
> Thank you for recognizing the practicality of our proposed method! We have addressed your questions and conducted additional experiments to help you better evaluate our work.
>
> ---
>
> > Figure 1 does not fully illustrate the issue that parallel token sampling within a scale may lead to structural errors. Because this structural error is caused by replacing the token maps, which merely indicates that errors at the early scales are earlier to affect the generation quality of the model.
>
> Figure 1 primarily illustrates the importance of early stages in VAR. However, the parallel sampling mechanism may lead to unreasonable token maps in early stages, thereby causing structural errors in the generated images. This is an inference rather than a direct representation in Figure 1. Examples of structurally flawed images generated by VAR are shown in Figures 4, 8, and 9.
>
> > Increasing the sampling steps in each scale is a more fundamental and direct method, which directly addresses the essence of the issue. However, there were no corresponding comparative experiments in the paper.
>
> **This would violate the design philosophy of next-scale prediction in VAR models.** One of the core design principles of VAR is intra-scale parallel sampling (even though this approach has its issues), which is completed in a single step. Increasing the number of sampling steps within each scale would cause VAR to degenerate into a traditional autoregressive image generation model based on next-token prediction.
>
> The reason VAR achieves extremely fast generation is that it performs autoregressive generation only across scales, while sampling all tokens in parallel within each scale. Adding sampling steps inside each scale would not only require extensive modifications to the VAR model itself and retraining the model, but would also significantly increase inference cost. Most importantly, this inherently violates the design philosophy of VAR’s next-scale prediction. In contrast, LSRS does not interfere with VAR itself; it only performs filtering after sampling, making it completely transparent to the VAR model.
>
>
> > Why is it said that applying LSRS from the first scale causes many samples to converge to similar values at this scale in line 370? There is no direct evidence in the paper.
>
> We thank the reviewer for this insightful observation. In fact, our conclusion is drawn from a joint analysis of the generation metrics (Table 3), the scoring model performance (Figure 5), and the mechanism of LSRS:
>
> * Since the scoring model **almost fails at the first scale**, this immediately makes $ST=1$ a poor choice. As shown in Figure 5 (top right), the validation accuracy of the scoring model at scale 1 is only about 52%, close to random guessing. This indicates that at this scale, the scoring model functions merely as a biased filter rather than an effective quality evaluator.
>
> * **Table 3 clearly shows that under the $ST=1$ setting, as the number of candidates $M$ increases, the FID score improves slightly at first but then rapidly deteriorates.** The first scale contains only a **single token**, which essentially acts as a **global bias** applied to all tokens. When $M$ becomes too large, selecting the highest-scoring token from the candidate pool causes the model to **overfit** the scoring network’s bias. This collapses the generation distribution toward the scoring model’s preferred subset. The resulting loss of diversity leads to the **FID deterioration** observed in Table 3.
>
> Nevertheless, our main point is that, due to the unique characteristics of the first scale, we recommend starting the application of LSRS from the second scale.

---

> ### Author Response · Authors · 2025-11-16
>
> > The ratio between computational cost and performance return is not adequately discussed in the later scales, because it is observed that earlier scales have a greater impact on the image structure in the paper.
> >
> > Why are no ablation experiments conducted on the later scales about the use or non-use of LSRS?
>
> Although the experimental results in Figure 3 already demonstrate the importance of early scales, we agree that adding ablation experiments that apply LSRS only at individual scales would make the conclusion more intuitive. **The additional experimental results are as follows:**
>
> | Scale  | FID (A)  | FID (B)   |
> |-----|-------|-------|
> | -   | 1.95  | 1.95 |
> | 1   | 2.30  | 2.49 |
> | 2   | 1.68  | 1.80 |
> | 3   | 1.79  | 1.88 |
> | 4   | 1.82  | 1.90 |
> | 5   | 1.90  | 1.93 |
> | 6   | 1.91  | 1.93 |
> | 7   | 1.92  | 1.94 |
> | 8   | 1.93  | 1.94 |
> | 9   | 1.95  | 1.95 |
> | 10  | 1.95  | 1.95 |
>
> The experiments are conducted on VAR-$d30$ with $M=32$ fixed. Here, FID (A) denotes applying LSRS starting from a given scale and continuing for all subsequent scales, while FID (B) denotes applying LSRS only at that specific scale and not at others. From the results, we can see that using LSRS at early scales is indeed more beneficial. Moreover, FID (A) consistently outperforms FID (B), with a larger advantage at early scales. Therefore, LSRS needs to be applied as early as possible and also maintained at later scales in order to achieve the best performance.
>
>
> > Can enhancing the ability of the score model improve the generation performance?
>
> **Yes, and this is almost inevitable.** However, the capability of our current scoring model has already nearly reached its upper limit, which is what enables the results presented in the paper. We demonstrate this point indirectly by weakening the scoring model. We train the scoring model using smaller portions of data ( 20%, 40%, 60%, 80% ) and use these weakened scoring models for sampling in LSRS. The results are as follows:
>
> | Training Data Percentage | val_loss | val_acc | FID  |
> |--------------------------|----------|---------|------|
> | Original VAR-$d30$    | -     | -    | 1.95 |
> | 20%                      | 0.574    | 70.4%   | 1.84 |
> | 40%                      | 0.513    | 74.7%   | 1.79 |
> | 60%                      | 0.483    | 76.5%   | 1.76 |
> | 80%                      | 0.446    | 78.3%   | 1.72 |
> | 100%                     | 0.424    | 79.7%   | 1.68 |
>
> The experiments are conducted on VAR-$d30$ with $ST=2, M=32$ fixed. We can see that as the scoring model’s capability improves, the generation quality also improves. This confirms the point.

---

> ### Author Response · Authors · 2025-11-25
>
> Dear Reviewer Wg8z
>
> Thank you very much for your detailed review comments. We have addressed your concerns in our rebuttal and included additional experiments. We hope you can find some time to review our responses. Should you have any further questions, please feel free to let us know.
>
> Best
>
> Authors

---

### Official Review · Reviewer_QqZ6 · 2025-11-01

**Soundness:** 2
**Presentation:** 2
**Contribution:** 2
**Rating:** 4
**Confidence:** 4

**Summary:**

The authors propose Latent Scale Rejection Sampling for VAR, which trains a lightweight scoring model to evaluate multiple candidate token maps and select the most high-quality one to guide subsequent generation. Experimental results demonstrate that this method improves generation quality while incurring negligible inference latency.

**Strengths:**

1. The paper is well-written and easy to follow.
2. The experiments and ablation studies are detailed and comprehensive.

**Weaknesses:**

1. The paper does not discuss the generality of the proposed method. Since all experiments are conducted on ImageNet, the scoring model is also trained on ImageNet, which is in-domain data. Can a scoring model trained on ImageNet be effectively applied to other domains, such as text-to-image models? Or, to adapt this method for text-to-image generation, would it be necessary to retrain the scoring model on domain-specific data?
2. The structural error problem in parallel decoding is not first identified in this paper. The related work Infinity[1] has also discussed this issue and proposed a training-based solution to address it. Since rejection sampling cannot inherently eliminate structural errors but only select relatively better outputs, how does the proposed method perform in comparison with Infinity?
3. How does the proposed method perform on relatively smaller models? In Table 1, results are only reported for the FlexVAR 1B and VAR 2B models, while no results are provided for smaller models such as VAR 300M or VAR 600M.

[1] Infinity: Scaling Bitwise AutoRegressive Modeling for High-Resolution Image Synthesis

**Questions:**

Please refer to the Weakness section.

---

> ### Author Response · Authors · 2025-11-15
>
> Thank you for considering our paper well-written and the experiments thorough! We have addressed your questions in detail and clarified some misunderstandings and confusions. We hope our responses further demonstrate the value of this research：
>
> ---
>
> > Can a scoring model trained on ImageNet be effectively applied to other domains, such as text-to-image models?
>
> **It cannot be directly applied to other models.** Our scoring model is currently trained only on ImageNet (which consists of real-world images), whereas text-to-image models are typically trained on much larger datasets that include a mix of various other styles of images. In those datasets, ImageNet may only represent a very small subset. Additionally, the specific implementation of LSRS varies for different models. For example, if LSRS is to be used with text-to-image models, the conditional input is no longer a category label but rather a text embedding, which requires the integration of an additional text encoder.
>
> > To adapt this method for text-to-image generation, would it be necessary to retrain the scoring model on domain-specific data?
>
> **Yes.** The most recommended approach is to strictly use domain-specific data to train the scoring model.
>
> > The structural error problem in parallel decoding is not first identified in this paper. The related work Infinity[1] has also discussed this issue and proposed a training-based solution to address it. Since rejection sampling cannot inherently eliminate structural errors but only select relatively better outputs, how does the proposed method perform in comparison with Infinity?
>
> **The Infinity paper does not seem to mention the structural error issue in the VAR model's parallel decoding.** It only addresses the **train-test discrepancy** caused by the **teacher-forcing training** method of VAR and proposes the Bitwise Self-Correction (BSC) to mitigate it. You may have been confused by lines 187-189 in our paper. However, our intention in those lines was to emphasize that if an unreasonable token map is generated due to **parallel sampling** at an early scale, the error will be amplified in subsequent scales because of error accumulation in autoregressive models.
>
> Therefore, our work consistently highlights the problems caused by the **parallel decoding** of the VAR model, while Infinity's BSC focuses on the **train-test discrepancy** issue in autoregressive models. The two approaches aim to solve **different problems** and cannot be directly compared.
>
> > How does the proposed method perform on relatively smaller models? In Table 1, results are only reported for the FlexVAR 1B and VAR 2B models, while no results are provided for smaller models such as VAR 300M or VAR 600M.
>
> Actually, we explicitly stated at line 309 of our paper: **"More experimental results on the VAR model are provided in Appendix B."** This includes the results for both the VAR 300M and VAR 600M models. The results show that LSRS is effective for VAR models of varying scales.

---

> ### Author Response · Authors · 2025-11-25
>
> Dear Reviewer QqZ6
>
> Thank you very much for your detailed review comments. We have addressed your concerns and clarified the misunderstandings in our rebuttal. We hope you can take a moment to review our responses at your convenience. Should you have any further questions, please feel free to let us know.
>
> Best
>
> Authors

---

> > ### Comment · Reviewer_QqZ6 · 2025-11-28
> >
> > Thank you for the response. My concerns are not fully addressed.
> >
> > Regarding Weakness 1: this substantially limits the generality of the method, as strong performance depends on having access to in-domain data. However, for most publicly available text-to-image models, their training data are not accessible, which means the method would be considerably less effective in practice.
> >
> > Regarding Weakness 2: in the original VAR training setup, all context tokens are correct. Consequently, during inference, the parallelized independent sampling of tokens at the same scale can lead to structural errors. In your method, since the training procedure is unchanged, you attempt to post-hoc refine the parallel-generated tokens. In contrast, Infinity recognizes the same issue but addresses it directly during training by intentionally corrupting part of the context tokens. This ensures that, during inference, even if some parallel-generated tokens are incorrect, they still fall within the training distribution. Inherently, both approaches aim to resolve the same underlying problem.

---

> ### Author Response · Authors · 2025-11-28
>
> Thank you for taking the time to review our response. Regarding any remaining concerns you may have, we provide further clarification below. We hope this will help you better evaluate our work:
>
> ---
>
> > Regarding Weakness 1: this substantially limits the generality of the method, as strong performance depends on having access to in-domain data. However, for most publicly available text-to-image models, their training data are not accessible, which means the method would be considerably less effective in practice.
>
> We first summarize our previous response:
>
> * The scoring model trained on ImageNet cannot be used **directly** for text-to-image models, because the data distributions differ significantly (ImageNet is only a subset), and the scoring model needs to incorporate a text encoder.
> * We recommend strictly using domain-specific data to train the scoring model, as this is the best practice. However, **domain-specific data** is not equivalent to the **official training data**.
>
> Regarding the difficulty of obtaining the **official training data** for text-to-image models, researchers can simply replace it with **public datasets**. Our experiments focus on class-conditional generation (ImageNet), but the methodology of “training a discriminator to filter out structural errors” is **general**. For text-to-image models, researchers only need to apply the same training procedure using **available** image–text pairs collected on their own. This is **standard practice in the field**, consistent with all works that propose **new text-to-image models** and with all **training-based improvement methods**.
>
> > Regarding Weakness 2: in the original VAR training setup, all context tokens are correct. Consequently, during inference, the parallelized independent sampling of tokens at the same scale can lead to structural errors. In your method, since the training procedure is unchanged, you attempt to post-hoc refine the parallel-generated tokens. In contrast, Infinity recognizes the same issue but addresses it directly during training by intentionally corrupting part of the context tokens. This ensures that, during inference, even if some parallel-generated tokens are incorrect, they still fall within the training distribution. Inherently, both approaches aim to resolve the same underlying problem.
>
> We agree with your observation that both methods aim to reduce generation errors. However, it is important to point out that LSRS and Infinity target completely different error–formation mechanisms. The two are in fact **complementary and orthogonal**, rather than competing:
>
> * In our paper, we explicitly point out that VAR’s **within-scale parallel sampling** may lead to structural errors. Thus, we propose LSRS to implicitly model the dependencies among tokens within a scale, thereby reducing structural errors. The autoregressive error accumulation we later mention only **amplifies** this issue, and is not our main focus. LSRS is **entirely** designed for scenarios involving **multi-token parallel sampling**.
>
> * In contrast, Infinity has never discussed the potential issues caused by within-scale parallel sampling. Their focus is solely on the train–test discrepancy arising from **teacher-forcing training**, which exists in all autoregressive models. Importantly, although they propose Bitwise Self-Correction (BSC) to alleviate this, it still does not directly address the problems introduced by parallel sampling. Even if Infinity is tolerant of erroneous previously generated tokens during the model's **forward pass**, the tokens in the next scale are still generated via **parallel independent sampling**.

---

### Meta-Review · Area_Chair_RSyk · 2026-01-07

**Summary:**

All reviewers started with negative scores. Their early concerns were mainly about limited generality beyond ImageNet and the way the problem is framed. QqZ6 focused on text to image use and pointed out a key limit: the method needs an in-domain scoring model, which is not practical for modern closed models. Wg8z also argued that LSRS is a post hoc filter that screens outputs, rather than fixing the root cause of structural errors in sampling. The rebuttal did not give a principled answer to these points, so there is no basis for acceptance. We encourage the authors to address these limitations and resubmit to another venue.

**Reviewer Concerns:**

Addressed:

Compute: qV4B assumed LSRS reruns the full model many times. The authors clarified it samples latents without extra transformer passes, so the overhead is about 1 percent.

IS drop: qV4B asked why Inception Score drops. The authors explained it comes from safer samples.

Ablations: Wg8z asked for results on when to apply LSRS. The authors added numbers for later scales and for scale 1.

Outstanding:

Generality: QqZ6 still sees a major gap for text to image and broad use. Requiring an in-domain scoring model trained on the target distribution does not fit closed or hard-to-access models.

Novelty: Wg8z still views LSRS as selection, not a real fix. This keeps the contribution closer to a patch than a new solution.

**Reviewer Scores:**

QqZ6: likely 4. The generality issue remains.

Wg8z: likely 4. The main ad-hoc patch concern is unchanged.

qV4B: likely 5. The compute and IS points were addressed.

kcQM: likely 4. The formulation is still not clear to them.

---

### Decision · Program_Chairs · 2026-01-26

Reject